# Geography and availability of natural habitat determine whether cropland intensification or expansion is more detrimental to biodiversity

Silvia Ceaușu [1] ✉, David Leclère [2] & Tim Newbold [1]

To mitigate biodiversity loss from agriculture, intensification is often promoted as an alternative to farmland expansion. However, its local impacts remain debated. We assess globally the responses of three biodiversity metrics—species richness, total abundance and relative community abundance-weighted average range size (RCAR), a proxy for biotic homogenization—to land conversion and yield increases. Our models predict a median species loss of 11% in primary vegetation in modified landscapes, and of 25% and 40% in cropland within natural and modified landscapes, respectively. Land conversion also reduces abundance and increases biotic homogenization, with impacts varying by geographic region and history of human modification. However, increasing yields changes biodiversity as well, including in adjacent primary vegetation, with effects dependent on crop, region, biodiversity metric and natural habitat cover. Ultimately, neither expansion nor intensification consistently benefits biodiversity. Intensification has better species richness outcomes in 29%, 83%, 64% and 57% of maize, soybean, wheat and rice landscapes, respectively, whereas expansion performs better in the remaining areas. In terms of abundance and RCAR, both expansion and intensification can outperform the other depending on landscape. Therefore, minimizing local biodiversity loss requires a context-dependent balance between expansion and intensification, while avoiding expansion in unmodified landscapes.

Agriculture is one of the major drivers of biodiversity loss[1,2] and many projections suggest that biodiversity loss due to agriculture may continue to occur locally, even under ambitious scenarios that include efforts to limit future demand for agricultural products[3–5]. Policies attempting to reduce the biodiversity cost of agriculture often aim to reduce farmland expansion by increasing yields on existing agricultural land[6,7]. To understand the effectiveness of this strategy we need to fully quantify the biodiversity impact of yield increases and compare it to the impact of farmland expansion. Considering there is a large uncertainty about where land-use pressure could increase in the future[8], we also need to understand the relative biodiversity costs of expansion and intensification across global landscapes.

The intensification–expansion options are often discussed either in terms of spatial hotspots of potential conflict with biodiversity conservation[9], or within the land-sparing/sharing framework[10,11]. In the latter approach, the two ends of a continuum of options are extreme

[1]Centre for Biodiversity and Environment Research, Department of Genetics, Evolution and Environment, University College London, London, UK. [2]Biodiversity and Natural Resources (BNR) Program, International Institute for Applied Systems Analysis (IIASA), Laxenburg, Austria. ✉e-mail: s.ceausu@ucl.ac.uk

land sparing, when production is concentrated on the smallest possible area by intensifying agriculture, and extreme land sharing, when production is spread with the lowest possible local impact through the use of biodiversity-friendly practices[10]. The land-sparing/sharing studies often rely on models of population density in relation to yield at regional scales as a measure of likelihood of species persistence, aimed at understanding extinction risk. Most of these studies conclude that more species would benefit from land sparing than from land sharing at a given production target[7,12–14], whereas other studies suggest that species responses are mediated by surrounding natural habitat[15–18] and land-use history[19–21]. Studies of local community metrics in relation to yield, which is important to understand ecosystem service supply[22], and stability of ecological communities[23] are rarer, and show mixed responses of biodiversity to yield increases[24–27]. Neither species persistence nor community metrics responses to yield increases were assessed across global agricultural landscapes, so our understanding of the comparative biodiversity impacts of agricultural intensification and expansion is far from complete.

Quantifying land-use intensity globally is a complex challenge due to the multiple dimensions of intensity, and the gaps and uncertainty of many data sources[28]. To quantify the biodiversity impacts of agricultural management, previous large-scale studies have used output proxies such as human-appropriated net primary productivity[29], agricultural inputs such as fertilizers[29] or broad categories of land-use intensity, which typically consider both agricultural inputs and outputs[30,31]. As agricultural management is only a means to obtaining a certain level of yield, focusing instead on the biodiversity impact of yield provides a more direct relationship to consumption patterns, which foregrounds the importance of the demand-side measures to reducing biodiversity impacts of agriculture[4]. This approach also provides a more direct relationship to agricultural production, as higher agricultural inputs do not always translate into higher yield, and degraded farmland might actually require higher inputs for the same yield level[32]. Similarly, yields can be increased with lower biodiversity impacts when relying on ecosystem services[33,34] or using technological improvements benign for biodiversity[35].

To fully quantify the biodiversity impact of yield increases, we have to look beyond the most frequently used metrics of species richness[30,31,36] and total abundance[31,36], which can obscure changes in community composition. Biotic homogenization, which happens when ecological communities become increasingly similar across space[37], is often a consequence of human impact even when other biodiversity metrics appear unchanged[38]. Among species characteristics used to capture changes in community composition and biotic homogenization[39,40], range size is a key predictor of extinction risk[41] and therefore highly relevant for conservation goals. Community-average range size, often weighted by the abundance of the component species, reflects shifts in composition towards more geographically widespread or narrow-ranged species, indicating whether a community becomes more or less distinctive[42]. Apart from Phalan et al.[7,11], who report how global ranges of birds and trees in Ghana and India relate to species' ability to persist in agricultural landscapes at different yield levels, few studies report how biotic homogenization relates to agricultural management[42], and none to different yield levels at a global scale.

Here we provide a global assessment of the biodiversity consequences of land conversion and yield increases in landscapes producing four crops: maize, soybean, wheat and rice, which together represent over half of the total global calorie production[43]. We achieve this in four stages, each focused on a specific question: (1) what is the biodiversity impact of land conversion in the absence of any yield considerations, considering both local and landscape-level effects of land conversion? (2) Within agricultural landscapes, what is the biodiversity impact of increasing yields? (3) What is the biodiversity impact of reaching the maximum attainable yield in a given region, that is, closing yield gaps[6], in the absence of land conversion? And (4) how do the hypothetical impacts of land conversion and yield increases targeting an equal increase in production compare within each existing agricultural landscape, defined here as the extent of one raster cell with any amount of cropland? We analyse crops separately to capture potential differences in landscape types and landscape effects of different management approaches, and we focus on two land uses: cropland (that is, land used for cultivation of herbaceous crops) or primary vegetation (that is, land showing no evidence of prior destruction of vegetation, either by human actions or extreme natural events). We use a space-for-time substitution to model three biodiversity metrics: sampled species richness, sampled total abundance and relative community abundance-weighted average range size (RCAR). First, we model how biodiversity reacts to land-use conversion locally (farmland versus primary vegetation) and at landscape scale (modified landscapes with less than 30% natural habitat versus natural landscapes with more than 70% natural habitat) by creating a combined land-use–landscape variable (Fig. 1a,c). We chose these thresholds of natural habitat based on research showing that ~30% and ~70% mark important thresholds in how habitat loss and spatial arrangement of habitat patches interact to reduce biodiversity[44]. These values also broadly align with habitat cover thresholds identified for biodiversity in tropical forests[45] and in agricultural landscapes[46]. Second, we model biodiversity impacts of agricultural yield[47] and amount of natural vegetation in the landscape[48] (Fig. 1b,c). Relying on these models, we then answer the third and fourth questions (Fig. 1c). We first project the impact of fully closing yield gaps on existing cropland identified as having yields below the maximum attainable yields for the respective region and crop[6]. We then compare estimated biodiversity impacts of achieving an arbitrary increase in production by either expansion or intensification for each crop and within each raster cell already containing the respective crop, regardless of the yield levels in the year 2000, in accordance with the principle of comparing alternatives that are matched in terms of total production[10,49]. Although we test several production increase targets, we focus on the results for a 1% uniform production increase to minimize the number of raster cells for which we would achieve invalid land-use proportions in the expansion scenario, for example, cropland occupying more than 100% of the cell or primary vegetation contracting to below 0%. Our unrealistic scenario of 1% uniform production increase is obviously unsuitable to predict credible land-use change patterns, but is suitable to compare impacts of hypothetical intensification and expansion within each agricultural landscape. Given that most comparisons of intensification and expansion focus on conversion of primary or natural vegetation into cropland[11,17,50], our expansion projections also focus on the primary vegetation to cropland conversion. Moreover, stress testing the prevailing opinion that intensification is less detrimental to biodiversity than expansion requires a comparison of intensification against the most impactful type of expansion, which is the one at the cost of primary vegetation.

## Results

### Biodiversity impacts of habitat conversion

Relative to primary vegetation in natural landscapes, the other land-use–landscape categories showed reductions in species richness (Fig. 2a) and abundance (Fig. 2b,d), and increases in RCAR (Fig. 2c,e), although the statistically significant interactions of the combined land-use–landscape variable with other model variables suggest distinct patterns for different types of landscape. For example, the interaction with geographic region was selected as significant for total abundance and RCAR (Fig. 2b–e). The results suggest that in tropical regions, primary vegetation in modified landscapes remains relatively close in terms of abundance and RCAR to primary vegetation in natural landscapes, and the 95% confidence intervals include 0, indicating no significant difference between these two categories. In non-tropical regions, it is the cropland in natural landscapes that has biodiversity levels closest to primary vegetation in natural landscapes in terms

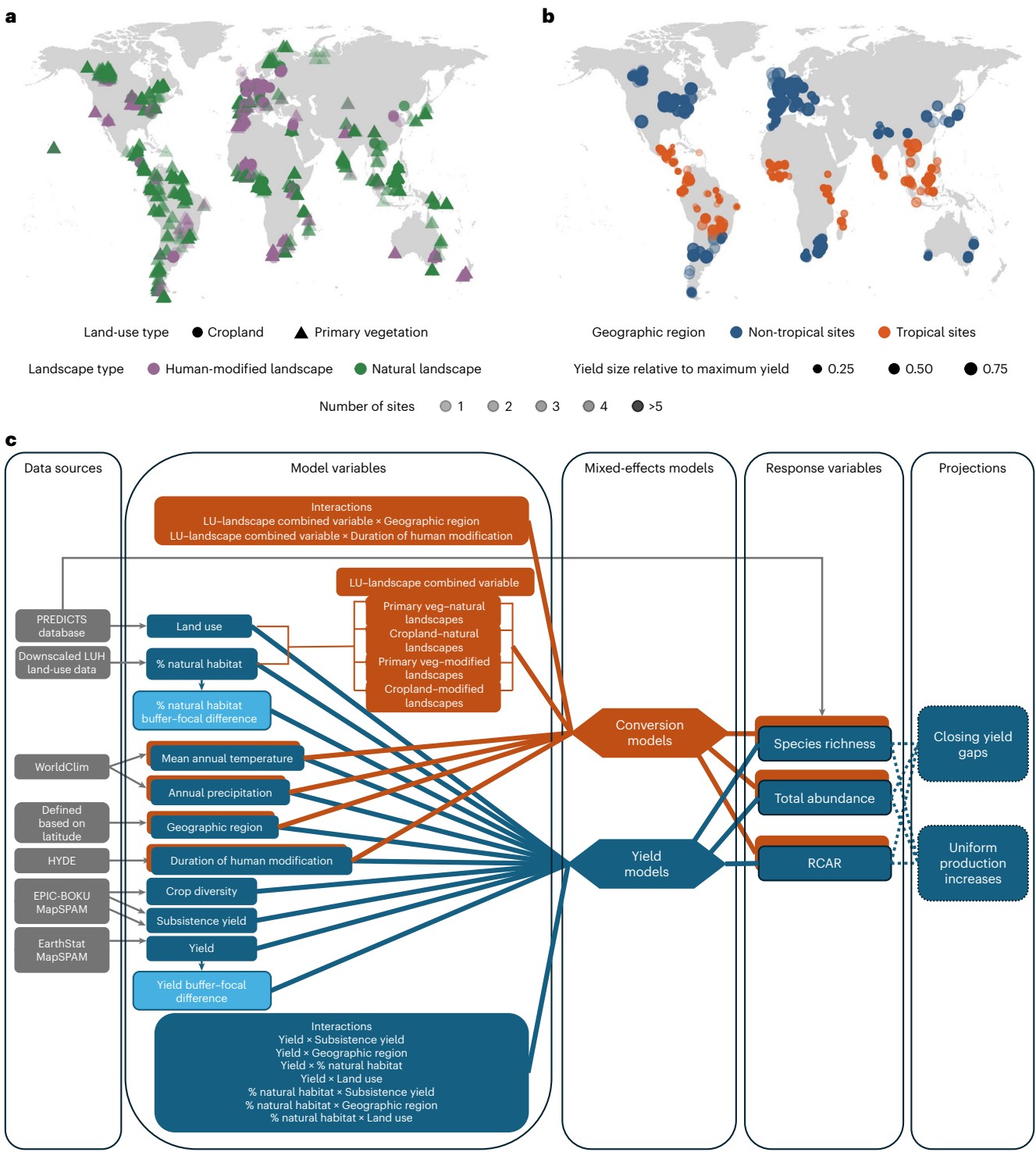

**Fig. 1 | Global distribution of PREDICTS sites used for the land conversion and yield–biodiversity models, and overview of data sources, variables, models and projections. a**, PREDICTS site distribution for the land-conversion models of biodiversity response to land use and landscape type. Circles and triangles show cropland and primary vegetation sites, respectively, and purple and green show human-modified (less than 30% of natural vegetation) and natural landscapes (more than 70% natural vegetation), respectively. **b**, PREDICTS site distribution for the yield–biodiversity models of biodiversity response to different yield levels of maize, soy, wheat and rice. Orange and blue indicate tropical and non-tropical sites, respectively, and circle size is proportional to the yield relative to maximum yield for each crop, aggregated across maize, soybean, wheat and rice. The relative yields were aggregated by weighting each crop yield by the size of the cropland for that particular crop out of the total agricultural area for

the four crops in each location. The symbols for PREDICTS sites in **a** and **b** are transparent to make overlapping sites visible. Please note that the transparency of symbols is reduced by overlap with sites that are in close proximity, not at the same exact location. **c**, Graphical illustration of the data sources, model variables and their use across the two types of models and projections. 'Yield' represents the landscape yield for each of the four crops, '% natural habitat' represents the percentage of natural habitat in the landscape, and 'yield buffer–focal difference' and '% natural habitat buffer–focal difference' represent the difference between the one-cell buffer value and the focal cell value for yield and percentage of natural habitat, respectively. 'LU–landscape' combined variables refer to the land-use–landscape combined variable. All other variable names are self-explanatory. Panels **a** and **b** made with Natural Earth world basemap.

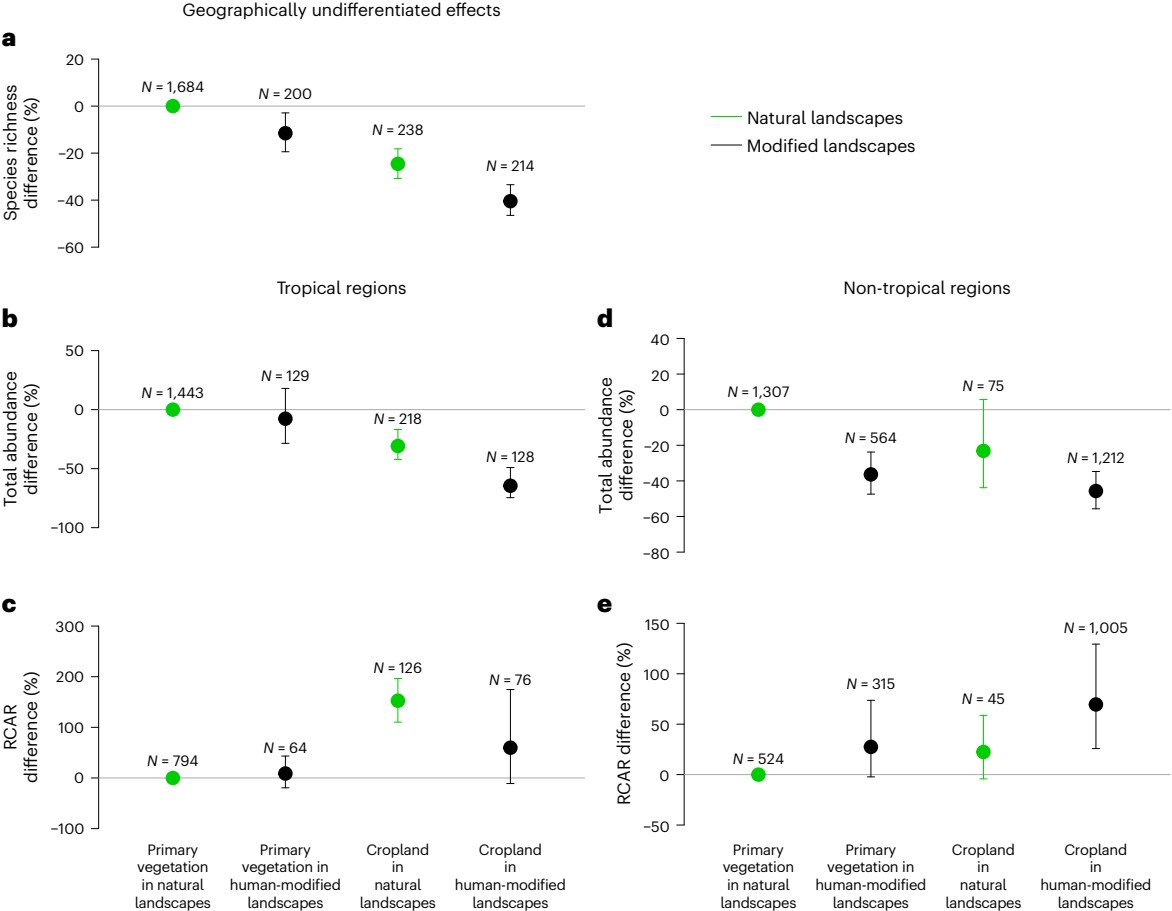

**Fig. 2 | The modelled effect of land conversion at local and landscape scale on three biodiversity metrics in tropical and non-tropical regions. a–e**, Modelled species richness (**a**), total abundance (**b,d**) and RCAR (**c,e**) in primary vegetation and cropland, in natural landscapes (>70% natural habitat, green dots and error bars) and in human-modified landscapes (<30% natural habitat, black dots and error bars). Species richness effects are geographically undifferentiated (the interactions between land conversion and geographic region are not statistically significant) whereas total abundance and RCAR have distinct tropical (**b,c**) and non-tropical effects (**d,e**). Models are based on PREDICTS data sampled in primary vegetation and cropland. The effects are presented as percentage changes relative to values in primary vegetation in natural landscapes. Median estimated values (points), and 2.5th and 97.5th percentiles (error bars), were calculated by sampling the fixed effects of the conversion models 1,000 times based on the variance–covariance matrix. If the error bar does not cross the 0% change line, this indicates that the difference between that particular class and the reference category is significant. The $N$ values represent the number of sampled sites for each class of the combined land-use–landscape variable for either tropical or non-tropical areas.

of total abundance (Fig. 2d) and RCAR (Fig. 2e). The species richness effects (Fig. 2a) and the non-significant interaction between land-use–landscape categories and geographic region for this metric suggests that species are lost with land conversion at similar rates in both tropical and non-tropical landscapes. It is important to highlight the imbalance in the number of data points between the different categories of the combined land-use–landscape variable and between data points in tropical and non-tropical regions, which might contribute to wider confidence intervals in some cases (Fig. 2 and Supplementary Table 1).

Other significant effects in the conversion models included mean annual temperature in the abundance and species richness models, and annual precipitation in the species richness model. Duration of substantial human landscape modification and its interaction with the combined land-use–landscape variable were retained in the species richness (Extended Data Fig. 1) and RCAR models (Extended Data Fig. 2). Areas that have experienced late substantial human landscape modification (in the past 500 years) exhibit more pronounced richness and RCAR differences across the land-use–landscape categories than areas that have experienced early substantial human landscape modifications (~2,000 years ago; Extended Data Figs. 1 and 2). The exception to this pattern is represented by RCAR in cropland in natural landscapes, where the increase is greater for landscapes with a long

history of substantial human landscape modification, although the small number of data points reduces confidence in this result (Extended Data Fig. 2b).

## Biodiversity impacts of yield increases

Biodiversity change with yield increases is mediated by land use, geographic region and percentage of natural habitat in the landscape. Yield increases were associated with decreases in species richness (Fig. 3a,c,d) with two exceptions: landscapes with soybean cultivation, where soybean yield increases were not associated with any changes in species richness (Fig. 3b); and landscapes with rice cultivation and high percentage of natural vegetation, where rice yield increases were associated with increases in species richness (Fig. 3d). The association between yield increases and total abundance is more complex (Fig. 3e–h), the interaction with geographic region becoming important to explain abundance patterns within landscapes with maize, soybean and rice cultivation (Fig. 3e,f,h, respectively). The general pattern in these landscapes is that total abundance increases with yield in tropical landscapes and decreases in temperate landscapes, while high levels of natural vegetation accentuate increases and dampen decreases in tropical and non-tropical landscapes, respectively. In landscapes used to grow wheat, total abundance decreases with increases in yield, most

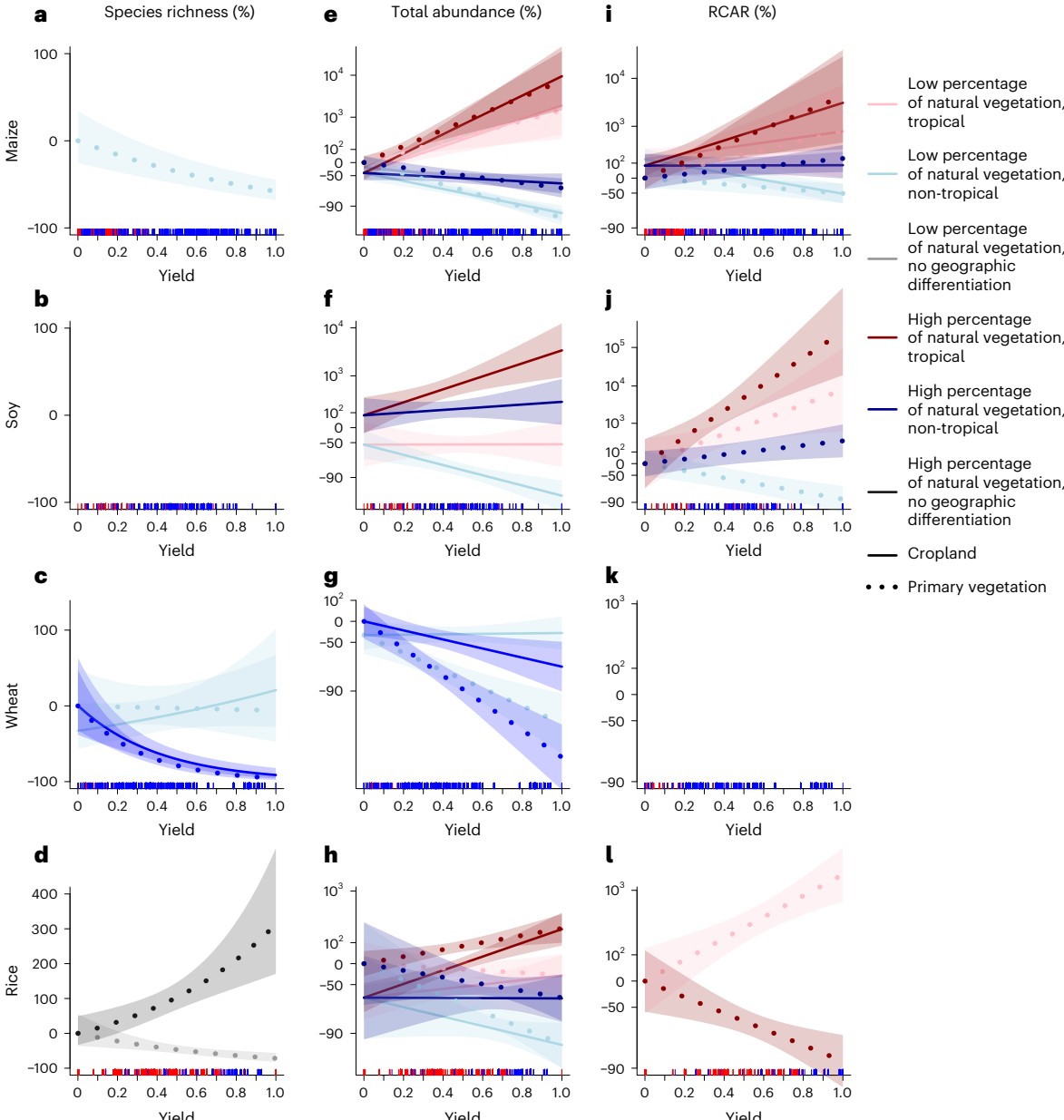

**Fig. 3 | The modelled effect of yield of four crops on three biodiversity metrics.** **a**–**l**, Modelled responses of species richness (**a**–**d**), total abundance (**e**–**h**) and RCAR (**i**–**l**) to maize (**a,e,i**), soy (**b,f,j**), wheat (**c,g,k**) and rice (**d,h,l**) yield. Where yield was not selected in the final model, the plot is empty. The interactions, if selected in the final model, are illustrated as follows: for yield × geographic region, red and blue colours represent tropical and non-tropical regions, respectively; for yield × percentage of natural habitat, light and dark colours represent trends at low and high percentages of natural habitat (15% and 85% for illustrative purposes), respectively; and for yield × land use type, solid and dotted lines represent cropland and primary vegetation, respectively. Where geographic region was not statistically significant either independently or in interactions, the trend line is grey. Where geographic region and land-use type were selected in the final model without their interactions with yield, we plotted the values for the category with the most data points. Where percentage of natural vegetation

was selected in the final model without its interaction with yield, we plotted the values for low percentage of natural vegetation. The lines represent median predicted values and shaded areas represent 95% confidence intervals. Species richness and total abundance increases are usually associated with positive, while RCAR increases are associated with negative biodiversity changes (that is, a homogenization of community composition). The ticks at the bottom of plots illustrate yield values rescaled to [0, 1]. The trends illustrate model predictions that do not necessarily represent plausible combinations of variable values. We scaled biodiversity metrics relative to the values in primary vegetation in landscapes with the lowest yield, a value of 0% meaning that biodiversity has not changed compared to this reference value. Owing to the large range of values, the changes in total abundance (**e**–**h**) and RCAR (**i**–**l**) were $\log_{10}$-transformed to facilitate illustration, and the *y* scale is logarithmic.

strongly in landscapes with a high percentage of natural vegetation (Fig. 3g). Geographic region impacts the effects of maize and soybean yield on RCAR, with tropical landscapes experiencing strong negative biodiversity impacts of increasing yields (that is, steep increases in RCAR), and non-tropical landscapes showing smaller impacts or even

decreasing RCAR with increasing yields (Fig. 3i,j). In landscapes with rice cultivation, RCAR decreases with increasing yield at high percentage of natural vegetation, but increases with increasing yield at low percentage of natural vegetation (Fig. 3l). Wheat yield is not associated with any changes in RCAR.

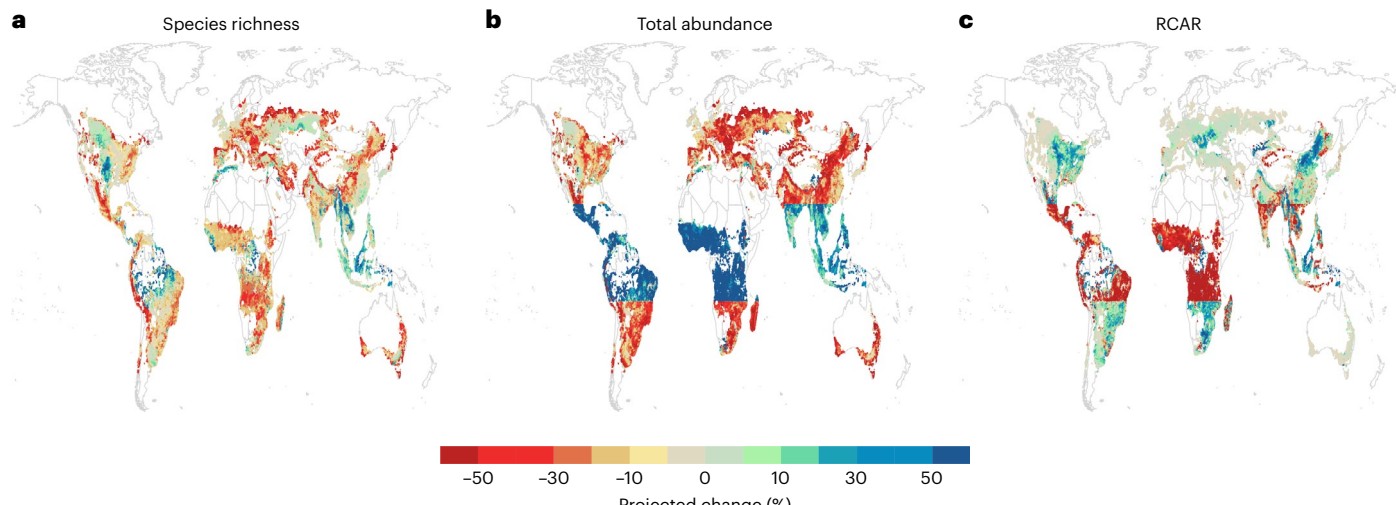

**Fig. 4 | The projected effect of closing yield gaps on three biodiversity metrics.** **a**–**c**, Local species richness (**a**), total abundance (**b**) and RCAR (**c**), calculated as the percentage difference between biodiversity at yield levels equal to those where yield gaps estimated to have existed in 2000 have been closed and biodiversity at yield levels estimated for the year 2000. The colour range symbolizes positive change (cold colours) and negative change (warm colours) for all plots, including for RCAR percentage changes, which were multiplied by −1. Note that yield benefits from closing yield gaps are spatially variable and depend on the size of yield gaps (Supplementary Fig. 23) and the cultivated area in each pixel (Supplementary Fig. 22). Figure made with Natural Earth world basemap.

The datasets were imbalanced in terms of number of data points available for the different land uses and geographic regions (Supplementary Figs. 1–4 and Supplementary Tables 2–5). The models were robust to the choice of subsistence yield and fairly robust to the choice of yield data (Supplementary Tables 9–19). Most models using yields from MapSPAM agreed with the models that used yields from Earth-Stat in direction, or in both direction and magnitude of biodiversity change (Supplementary Table 9). The only MapSPAM yield models that did not agree on the direction of the effect of yield on biodiversity were rice total abundance and maize RCAR models (Supplementary Tables 16 and 17).

Among the other variables, percentage of natural vegetation and land use were selected in all 12 best models (4 crops × 3 biodiversity metrics); geographic region, as yield, was selected in 10 models; and subsistence yield was selected in 8 models (Supplementary Table 8). When selected in the final model, subsistence yield had mostly negative effects on species richness and abundance. It also had a negative effect on RCAR (that is, positive impact in terms of biodiversity value) for all crops. All the other variables were selected in 4 or less of the 12 models. Crop diversity had a negative effect on species richness and abundance in maize landscapes but it had a positive effect on abundance in soy landscapes. Duration of substantial human landscape modification had a negative impact on total abundance in maize and rice landscapes, and a positive effect on RCAR (that is, negative impact in terms of biodiversity value) in soy landscapes.

The variance explained by fixed effects (marginal pseudo-$R^2$)[51] for all models varied between 0.05 and 0.35, whereas the variance explained by fixed and random effects together (conditional pseudo-$R^2$)[51] varied between 0.5 and 0.98 (Supplementary Tables 10–19). These levels of pseudo-$R^2$ are typical for models based on datasets collected from multiple sources, in which most of the variation is explained by the random effects[52]. The model diagnostic plots suggest that our models do not fully fit model assumptions (Supplementary Figs. 5–21). The use of the negative binomial instead of the Poisson distribution for the species richness models did not substantially improve model behaviour (Supplementary Figs. 5–12). The estimated effect sizes and directions were largely similar to the Poisson models (Supplementary Fig. 15 and Supplementary Table 20). The Bayesian models produced very similar results to the original models (Supplementary Table 21).

## Biodiversity impacts of closing yield gaps
When closing yield gaps for all four crops, as estimated for the year 2000, the biodiversity effects differ strongly across metrics and geographic regions. For non-tropical regions, closing yield gaps leads to negative species richness and abundance effects but positive effects in terms of RCAR (Fig. 4). For tropical regions, the effects in terms of species richness are both positive and negative depending on location, whereas the effects on total abundance are positive and the effects on RCAR are negative (Fig. 4).

More specifically, when closing yield gaps 73.6% of all grid cells would experience a decrease in species richness. Mean species richness change per pixel globally was −9% and the median was −8.7%. The decrease in species richness is spread across both tropical and non-tropical areas, while the increase in species richness is concentrated mainly in tropical areas (Fig. 4a), probably due to positive rice yield effects in landscapes with a high percentage of natural habitat (Supplementary Fig. 29). A total of 61.5% of all grid cells will experience a decrease in total abundance, with an average of +37% change driven by high abundance increases for maize and soybean in tropical areas (Supplementary Fig. 30), and a median of −10.6%. Non-tropical areas are almost all characterized by decreases in abundances (Fig. 4b). For RCAR, 38.4% of grid cells will experience a negative biodiversity impact and almost all of them are in tropical areas (Fig. 4c). The difference in geographic patterns are most probably driven by soybean and maize yield RCAR effects (Fig. 3i,j). The average projected RCAR change across all grid cells was +68.6% with a median of 0%. A total of 15.5% of grid cells had a 0% change in RCAR, these being mostly non-tropical landscapes with wheat cultivation (Supplementary Fig. 31).

## Farmland expansion versus intensification
When comparing hypothetical options for increasing total production by 1% in existing agricultural landscapes, neither expansion nor intensification had better estimated biodiversity outcomes in all locations (Fig. 5). Intensification was associated with better biodiversity outcomes in terms of species richness than farmland expansion on 29%, 83.3%, 64.2% and 56.7% of the cultivated areas for maize, soybean, wheat and rice, respectively (Fig. 5a–d). Farmland expansion was associated with higher species richness on the rest of the cultivated areas. In terms of total abundance, modelled intensification was associated

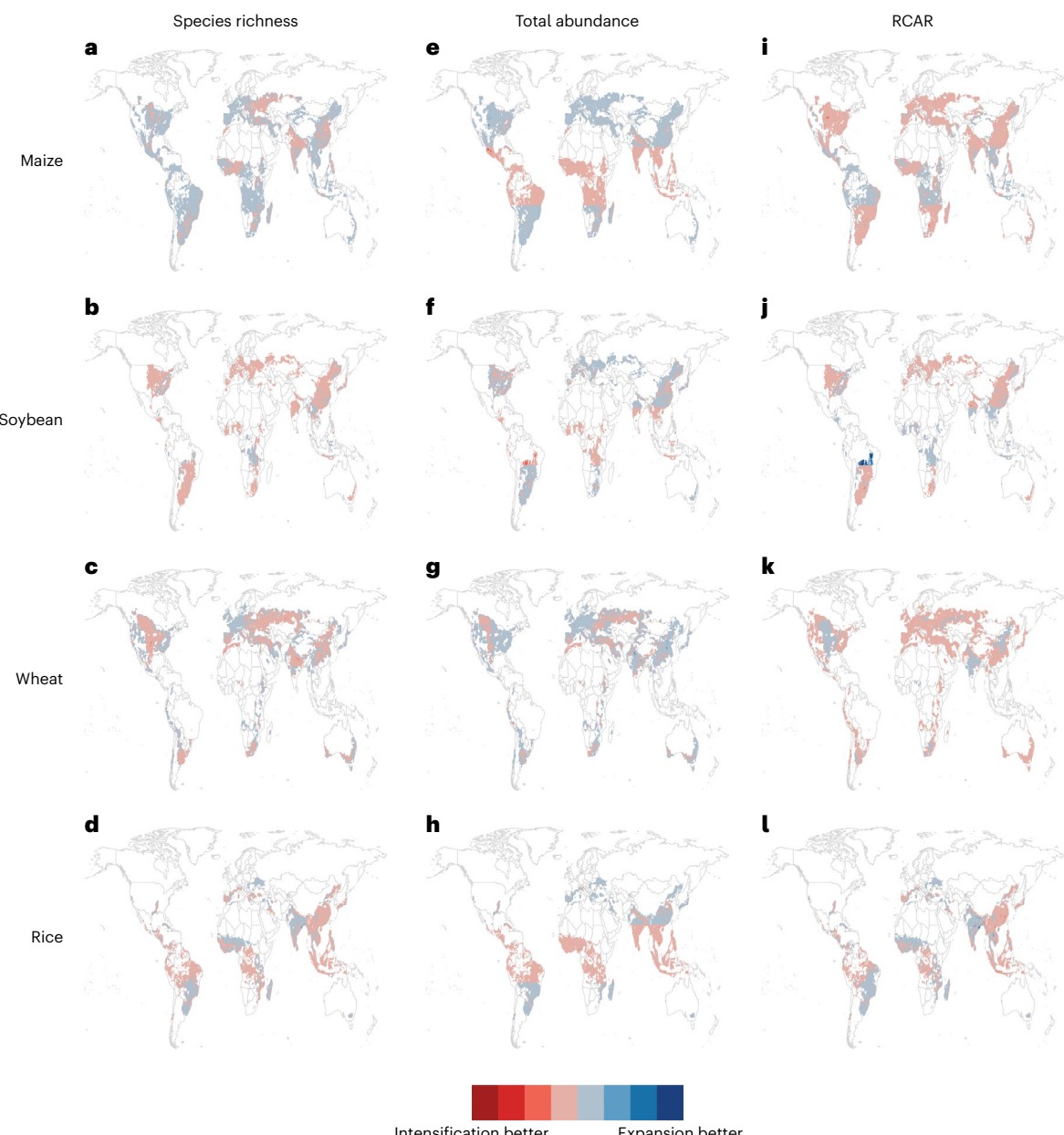

**Fig. 5 | The difference in biodiversity metrics when comparing land expansion and intensification within the same agricultural landscape. a–l,** The plots are organized in rows for each crop: maize (**a,e,i**), soybean (**b,f,j**), wheat (**c,g,k**) and rice (**d,h,l**). Columns represent each biodiversity metric: species richness (**a–d**); total abundance (**e–h**) and RCAR (**i–l**). The colour range symbolizes areas where increasing total production by 1% through cropland expansion is better for biodiversity (blue colours) and areas where increasing total production

by 1% through intensification is better for biodiversity (red colours). RCAR (**i–l**) increases and decreases were considered to be the negative and positive outcomes for biodiversity, respectively. We removed 0.9%, 2.3%, 0.5% and 0.2% of raster cells from the maize, soybean, wheat and rice analyses, respectively, owing to invalid projections of land-use coverage (that is, more than 100% cropland coverage or negative coverage of primary vegetation). Figure made with Natural Earth world basemap.

with more positive outcomes on 41.4%, 36.1%, 34% and 63% of the cultivated areas for maize, soybean, wheat and rice, respectively. Thus, farmland expansion was associated with higher total abundances on more than half of the land cultivated with maize, soybean and wheat, but intensification was estimated to be better across much of the tropics (Fig. 5e–h). For RCAR, farmland intensification resulted in better modelled outcomes on 73.7%, 71.1%, 71.6% and 48.3% of the cultivated areas for maize, soybean, wheat and rice, respectively. In this case, intensification was the better production-growing strategy compared with farmland expansion for three of the four crops: maize, soybean and wheat. However, farmland expansion resulted in better modelled outcomes in the tropics, especially for maize and soybean (Fig. 5i–l).

These overall patterns were similar when choosing different production increase targets (Supplementary Figs. 32 and 35) or when assuming different intensification patterns (Supplementary Figs. 33 and 34), including when 'restoring' some of the cropland to primary vegetation (Supplementary Fig. 36). The magnitude of the differences between intensification and expansion impacts was sensitive to the size of the production increase.

## Discussion

When looking at the biodiversity impact of land conversion separately from yield considerations, impacts of both local land use and landscape composition are mostly negative across species richness,

total abundance and RCAR, although their strength across land-use–landscape categories differs depending on geographic region and duration of substantial human landscape modification. Our results are consistent with previous studies of local land-use change[31,53,54] and landscape composition effects[55]. More specifically, our results highlight the importance of considering the landscape context when designing and implementing conservation management, as suggested by other authors[56]. For instance, biodiversity in primary vegetation in agricultural landscapes changes compared with primary vegetation in natural landscapes across all three metrics, especially in non-tropical areas. This suggests that primary vegetation in human-modified landscapes has less conservation value than primary vegetation in natural landscapes[57,58], and highlights the high biodiversity costs of expanding agriculture in previously unfarmed landscapes. Moreover, early human landscape modification is associated with lower biodiversity differences across most of the four land-use–landscape categories (Extended Data Figs. 1 and 2), consistent with previous research describing the biotic homogenization of regions with long histories of human disturbance[59,60]. However, primary vegetation in modified landscapes can harbour species of conservation concern[61] and probably represents an important source of ecosystem services for the surrounding farmland[55,62], especially when paired with favourable landscape configurations such as high density of edges between crop fields and non-crop areas[18].

When looking at the biodiversity impacts of yield or agricultural management, large-scale studies rarely assess how the direction of these effects might differ in different geographic regions (but see refs. 55,63). In our results, tropical landscapes cultivated with maize and soybean show an increase in total abundance and an increase in biotic homogenization, as measured by RCAR (Fig. 3), suggesting that abundance increases are driven by widespread species[42] and signalling a higher potential risk of invasion in these landscapes[64]. Indeed, previous research in tropical areas showed that species with large global ranges, which are often generalists[65], are more likely to persist and thrive in agricultural landscapes[7,11]. In non-tropical soy and maize landscapes, both total abundance and biotic homogenization generally decrease, suggesting that abundance decreases in these landscapes are also driven by changes in the abundance of widespread species, this time decreases. Supporting evidence for this pattern comes from abundance decreases in common European birds[66], often attributed to agricultural intensification[67]. There are two considerations that can help interpret these opposing responses of widespread species to yield increases. First, most tropical yields in our data are at the lower end of the yield range, whereas non-tropical yields are more equally distributed (Fig. 3, data distribution rugs at the bottom of all panels). Therefore, the tropical trends are likely to be representative of lower yields, whereas the non-tropical trends probably capture biodiversity trends across a wider yield range. Second, non-tropical areas, especially temperate regions, tend to have a longer history of human occupation and smaller areas of natural habitats than tropical landscapes[58]. This probably led to farmland-intolerant, narrow-ranged species being filtered out[59] and farmland-tolerant, widespread species increasing in abundance well before the accelerated intensification of the past decades. These ecological communities dominated by farmland-tolerant species in lower intensity agriculture still persist in eastern European traditional landscapes[68]. We speculate that the tropical biodiversity trends in our results capture the type of community change that had happened in the past in non-tropical agricultural landscapes, whereas the non-tropical trends capture mainly the response of farmland-tolerant species to the recent high levels of intensification in non-tropical areas.

Previous research has shown the importance of interactions between natural habitat and agriculture for mitigating biodiversity impacts[17,55], but our results suggest mixed impacts depending on crop and biodiversity metric. For example, in landscapes with wheat cultivation, areas with a higher percentage of natural habitat experience steeper decreases in species richness and abundance with yield, probably from higher initial biodiversity as natural habitat is positively related to species richness and abundance (Supplementary Table 20). This suggests that aspects of wheat landscapes or their management might undermine the ability of natural habitat patches to buffer the impact of yield increases, in accordance with research highlighting the importance of landscape context for the biodiversity value of natural habitat patches[69]. In maize and soybean landscapes, a higher percentage of natural habitat has a positive impact on the effect of yield on total abundance, accentuating increases in tropical and dampening decreases in non-tropical regions, but a negative biodiversity impact in terms of biotic homogenization, accentuating RCAR increases in tropical and dampening RCAR decreases in non-tropical regions. These results suggest that natural habitat in these landscapes is beneficial mostly to widespread species that are able to use natural habitat to benefit in relative terms from increases in yield[70]. In landscapes with rice cultivation, a higher percentage of natural habitat has a positive impact on the effect of yield on all biodiversity metrics, as species richness increases and RCAR decreases with yield at high percentages of natural habitat. In terms of total abundance, tropical increases become steeper and non-tropical decreases become shallower at high percentages of natural habitat. The RCAR responses to the interaction between rice yield and percentage of natural habitat suggest that narrow-ranged species are the ones benefiting in relative terms from yield increases in landscapes with a high percentage of natural habitat. We speculate that this could be owing to the specific requirements of rice cultivation that often involve a period of water submergence[71], which creates temporary wetlands that can be functionally equivalent to more natural wetlands[72]. Given that wetlands represent only 5–10% of the global land surface and that more than half of the global wetland area has been destroyed or modified[73], species that are likely to benefit from anthropogenic wetlands created by agriculture will be relatively narrow ranged[74]. Our results suggest that a certain amount of natural habitat is necessary for this to happen, as suggested by research that shows the importance of natural habitat for connecting populations of wetland species[75].

Although closing yield gaps was proposed more than a decade ago as a solution to the challenge of producing food while protecting biodiversity[76], there are surprisingly few quantitative assessments of its biodiversity impacts. Compared with existing assessments, our results suggest more severe losses in local biodiversity when closing yield gaps. Kehoe et al.[50], who analysed the biodiversity impact of maximizing agricultural intensity on all existing farmland, estimated a maximum of 7% loss in species richness and 13% loss in abundance from closing yield gaps. This contrasts with our projections, which suggest that almost 54% of cultivated grid cells will lose more species richness and 47% will lose more abundance than Kehoe et al.'s estimated maxima. Kehoe et al.[50] used modelled estimates of biodiversity loss from a global study of land-use impact on biodiversity, using the same dataset from which we derived our data[31]. The difference in results is probably owing to the focus of Newbold et al.[31] on agriculture in general rather than crop type, and the absence of interactions between intensity and natural habitat or geographic region in their statistical models[31]. As indicated by our analysis, these elements can have opposing effects on biodiversity, and therefore the negative impacts can be obscured in simpler models. Comparisons with existing assessments of the biodiversity costs of closing yield gaps is complicated further by the use of different biodiversity metrics. For example, Egli et al.[77] used a metric capturing the value of each pixel with any amount of farmland for the global survival of terrestrial mammals, birds and amphibians. Based on assumed responses of species persistence to agricultural intensification based on habitat preferences, the authors estimated a decrease of 11.1% in the global biodiversity value of agricultural lands due to closing yield gaps. Although the biodiversity metric used by Egli et al.[77] is not directly comparable to our metrics, the spatial patterns of

high biodiversity loss tend to agree, especially with our RCAR patterns in the tropics.

Considering the reduced biodiversity in primary vegetation in modified landscapes (Fig. 2) and the strong association between yield and negative biodiversity outcomes in some landscapes (Fig. 3), it is not surprising that agricultural intensification is not always the best solution for increasing agricultural production in terms of local biodiversity outcomes (Fig. 5), as suggested by other authors in the case of European agricultural landscapes[78]. In contrast, previous global analyses have emphasized the land-saving potential of agricultural intensification[79–81], which on its own is positive for biodiversity. Land-sparing/sharing studies at regional scale also usually conclude that land sparing, that is, intensification, results in better outcomes in terms of reducing extinction risks[10]. To the extent that biotic homogenization of communities as measured here does lead to higher extinction risks, our results disagree with the land-sparing/sharing research, as many tropical areas in our results experience higher biotic homogenization when raising yields than when expanding cropland at the expense of primary vegetation in agricultural landscapes. Unlike most land-sparing/sharing studies, we were not able to compare the biodiversity costs of extreme sharing and sparing, which could involve contraction of cropland and restoration of biodiversity, as that would involve additional modelling and assumptions regarding restoration approach and timeline of biodiversity recovery[82–84] (but see Supplementary Fig. 36 for a simplified partial restoration scenario). In our study, models also do not account for the configuration of cropland and primary vegetation areas in our pixels, which is an important aspect of the land-sparing/sharing framework, which recommends sharing or sparing of patches of at least 1–10 km$^2$ (ref. 10). Moreover, it is important to emphasize that we are comparing intensification and expansion within existing agricultural landscapes, and that expanding agriculture into new landscapes will lead to considerable decreases in biodiversity (Fig. 2).

Our study has several limitations that need to be considered when interpreting the results. First, our focus on the direct association between biodiversity metrics and yields means that relationships may change if agricultural management practices differ from those used at the sampled locations and times. As a consequence, our results are most representative of the time window represented by the EarthStat and Projecting Responses of Ecological Diversity in Changing Terrestrial Systems (PREDICTS) data, which is the time period between 2000 and 2005, and of the agricultural intensification type predominant during this time span, which relied on conventional approaches such as input increases. Although it is reasonable to assume that agricultural practices have not changed dramatically since then, closing yield gaps or sustainable intensification would probably result in considerable changes in management practices in some areas, which might change the biodiversity impacts presented here. The focus on yield will probably also obscure specific impacts of agricultural management that are not necessarily reflected in yields[85]. Moreover, our results are relevant for the set of crops considered here, and it is uncertain whether they would generalize to other crops. For example, agroforestry or multiannual cultivation systems might have different requirements for increasing yields[24,27], and these requirements may have different impacts on biodiversity from those presented here. Second, our cropland data were not sampled necessarily within cropland cultivated with the particular crops for which we are using landscape-scale yield estimates. Therefore, our models capture landscape-scale but not local-scale impacts of crop-specific management. In addition, our yield data might not capture effectively the intensity levels of the management of other crops in the landscape. We reduced the impact of this limitation by weighting our data points in the statistical analysis by the proportion of the respective crop in the landscape. Third, our choice of classifying sites into tropical or non-tropical regions based on latitude, although avoiding other limitations (Methods), leads to

unnatural artefacts in our figures where the border between positive and negative impacts in tropical and non-tropical areas is often artificially straight (for example, Figs. 4 and 5), whereas any changes in the relationship between biodiversity and yield in real landscapes would be more gradual. Fourth, all the limitations inherent to the datasets we use, such as geographical and taxonomical biases in PREDICTS[86], and data and downscaling uncertainties in EarthStat[47], are carried over to our own study. Fifth, there are several limitations of using spatial biodiversity data to infer changes through time, most importantly an inability to consider time lags in biotic changes and responses to environmental changes[87]. Sixth, some of our models were less robust to the choice of yield data source (Supplementary Table 9) and their results should be interpreted with caution. Finally, predicting biodiversity values when closing yield gaps relies on novel combinations of variable values, which are likely to go beyond those used for model fitting (and those present within real-world systems). In the case of models using log links or log-transformed variables, as our models are, this is likely to lead to particularly unrealistic extreme values[88], which we counteracted by focusing here on the broad patterns of biodiversity change.

Both land conversion and increasing agricultural yields have substantial biodiversity impacts. In particular, closing yield gaps will probably lead to a much higher biodiversity cost than previously estimated[50], and the increase in biotic homogenization with agricultural yields in the tropics is particularly worrying from a global biodiversity perspective. Given the mostly positive impact of natural habitat on biodiversity and ecosystem services, there is probably a balance that can be struck between intensification and expansion in agricultural landscapes, but this balance might change depending on geographic region, crop, conservation goals and remaining natural vegetation in the landscape. While avoiding expansion of agriculture in unmodified landscapes remains critical for global biodiversity, a reassessment of the approach of closing yield gaps might be necessary, especially in terms of defining safe intensification levels that can preserve functioning local communities. Moreover, agricultural management must consider other essential ecosystem services in addition to biodiversity, such as carbon storage and water provisioning[89]. Hopefully, our results will motivate a renewed focus on important management questions, such as how much natural habitat is necessary in agricultural landscapes to maintain biodiversity at sustainable levels[44,90], how to obtain yield increases with lower biodiversity impacts[43] and how to reduce the overall demand for agricultural products, which will avoid the hard choice between intensification and expansion[4].

## Methods
### Biodiversity data

For biodiversity data, we used the 2016 release of the PREDICTS database, which contains 3,250,404 biodiversity records, mostly sampled from 2000 to 2012, from 666 published studies[86,91]. Each study contains data sampled with the same method across a gradient of land use or land-use intensity. The data in each study are grouped into one or more spatial blocks, each containing data from one or more sites. Each site is attributed one of six predominant land-use classes based on the information provided in the original papers, or by the authors of those papers: primary vegetation, secondary vegetation, plantation forest, cropland, pasture and urban. For our purposes, we selected only those sites located in cropland or primary vegetation, resulting in 1,318,867 records from 10,094 sites from 489 studies. The dataset included 18,853 species of which there were 3,994 vertebrates, 6,693 invertebrates, 7,269 plants, 894 fungi and 3 protists. For each of the statistical analyses on relative biodiversity of human-modified and natural landscapes, henceforth land-conversion models, and biodiversity impact of yield, henceforth yield–biodiversity models, we used subsets of these data (Supplementary Tables 1–5 and Supplementary Figs. 1–4), which we selected based on the methods described in the section 'Data processing for statistical analysis'.

We used three metrics: sampled species richness, total sampled relative abundance and RCAR. Sampled species richness represented the count of all species sampled at a site, as identified by the authors of the original study (not necessarily always resolving taxonomic synonymy). Total sampled relative abundance was calculated as the sum of abundance measures for all taxa at a site, which was available for approximately 85% of all sites. Abundance was reported in a variety of measurements, such as individual counts (~87% of sites with abundance data), coverage or frequency of occurrence across plots (~10%), group or pair counts (~1%), abundance of animal signs (~1%) and biomass (~0.05%). In cases in which abundance measurements were sensitive to sampling effort and where that effort differed among sampled sites within a study (~1% of records), we corrected the raw abundance measure by dividing it by its relative effort as a proportion of the effort of the most sampled site within a study[31]. RCAR is a measure of how widely or narrowly the species in a community are distributed, on average. An increase in RCAR metrics indicates a shift towards more widely distributed species on average, which could be caused either by increases in the number or relative abundance of wide-ranging species and/or by decreases in narrow-ranging species. Similarly, a decrease in RCAR indicates a change towards more narrow-ranged species, on average, which could be caused by increases in narrow-ranged species and/or decreases in wide-ranging species. We used an RCAR metric as calculated in ref. 42 based on occurrence data from the Global Biodiversity Information Facility (GBIF) database (https://www.gbif.org) and extracted at 55 km × 55 km resolution. Modelled responses of RCAR to land use and land-use intensity have previously been shown to be robust to using different grid resolutions to calculate area or occupancy from GBIF records, and also to using alternative measures of range size[42].

## Model explanatory variables

**Global agricultural estimates.** We used two global agricultural datasets: EarthStat[47] for model selection and visualizations, and MapSPAM[92–96] for robustness testing. Although MapSPAM has the advantage that it covers three years (2000[93], 2005[94] and 2010[95]) instead of only one year (2000 for EarthStat), EarthStat provides information on the resolution (that is, administrative level of reporting) of the initial agricultural data, which impacts the quality of the spatial modelling[47], and which we used to discard spatial estimates based on data of lower quality. MapSPAM also used biophysical potential of crops as an assumption underlying the spatial disaggregation of agricultural data, which risked creating circularity in our statistical models, because we control for agricultural suitability (see section 'Subsistence yield as proxy for agricultural suitability').

*EarthStat.* EarthStat agricultural estimates include global agricultural estimates for 175 crops in 206 countries around the year 2000 at a spatial resolution of 5′ × 5′ (~10 km × 10 km at the Equator)[47,97]. Given the widely variable data quality and limited availability of proxies for agricultural suitability across crops, we selected four of the major crops for which estimates are likely to be more reliable[97]: maize, soybean, wheat and rice. We selected for our analysis the EarthStat yield estimates that relied on subnational-level data at one and two administrative levels below national level (top two data quality scores in the yield data quality spatial layer).

*MapSPAM.* MapSPAM agricultural estimates were created with the Spatial Production Allocation Model (SPAM) at a global resolution of 5′ × 5′ (~10 km x 10 km at the Equator). The SPAM modelling approach combines several types of data (for example, market access models, biophysical suitability, information on agricultural management) to create an informed spatial prior covering the territory over which the agricultural statistics are disaggregated[92]. Together with land-cover information, this spatial prior was used to allocate agricultural area and production.

**Percentage of natural habitat.** We included percentage of natural habitat in our models because research has indicated its importance independently[55] and in interaction with other variables[17] for determining biodiversity patterns. For each data point, we extracted the percentage of natural habitat from the land-use estimates of ref. 98, which were obtained by downscaling the Land-Use Harmonization dataset[48,99]. The year 2005 of the land-use dataset is within the sampling time span of the PREDICTS data and therefore a suitable data source. We projected primary and secondary vegetation proportional cover estimates onto a Behrmann equal-area projection and resampled them to the EarthStat resolution and extent. We then summed them to obtain an estimate for natural vegetation cover.

For the yield–biodiversity models, we used percentage of natural habitat as a continuous variable. For the land-conversion models, we used percentage of natural habitat to select sites that were located in pixels with less than 30% natural vegetation ('human-modified landscapes'), and pixels with more than 70% natural vegetation ('natural landscapes'; Supplementary Table 1). We then combined this landscape classification based on natural vegetation with the PREDICTS land-use classification to create a combined land-use–landscape variable with four categories: primary vegetation in natural landscapes, primary vegetation in human-modified landscapes, cropland in natural landscapes and cropland in human-modified landscapes.

**Land use.** In addition to assessing biodiversity within cropland, we were also interested in how agricultural management affects biodiversity in adjacent primary vegetation, which plays a key role in preserving biodiversity and ecosystem services[62] but is also impacted by nearby farming activities[100]. We used the land-use classification from the PREDICTS database to define whether a data point is in cropland or primary vegetation.

**Geographic region.** Environmental and socioeconomic differences between tropical and non-tropical areas can influence the biodiversity impacts of land-use change and intensity[55,101,102]. Data points located between 23.5° N and 23.5° S were identified as tropical, and otherwise as non-tropical.

**Subsistence yield as proxy for agricultural suitability.** The suitability for growing crops varies widely, resulting in large differences in inputs and biodiversity impacts at similar yield levels. Moreover, agricultural suitability can correlate positively with different biodiversity metrics because conditions that are favourable for crops are often also favourable for other species[103,104]. As a proxy for agricultural suitability, we used yields modelled by a version of the Environmental Policy Integrated Climate (EPIC) model[105], which was specifically adapted to estimate yields at different management intensities[106–108] by researchers at the University of Natural Resources and Life Sciences (BOKU) in Vienna, Austria. Specifically, we used EPIC-BOKU model simulations under a subsistence system without fertilization or irrigation at 0.5° × 0.5° resolution. Although termed subsistence yields in the literature and here, these estimates do not represent real-world subsistence yields, which typically involve some inputs and management. For analysis with the EarthStat yield estimates for the year 2000, we averaged EPIC-BOKU subsistence-yield estimates between the years 1997–2003. For MapSPAM models, we also used a five-year averaging window around the focus years (2000, 2005 and 2010) in order to have subsistence yield values consistent with those in the EarthStat models.

We tested for the robustness of our statistical models to choice of subsistence yield by replacing EPIC-BOKU estimates with MapSPAM subsistence estimates in both the EarthStat and MapSPAM models. MapSPAM estimates of subsistence yield were obtained through a combination of data and expert opinion[92,109].

**Other variables.** Crop diversity was found to have a positive effect on local biodiversity in some studies[110]. We calculated a Shannon crop diversity index based on all EarthStat estimates for all 175 crops:

$$H' = -\sum_{i=1}^{n} p_i \ln p_i \qquad (1)$$

where $H'$ is the Shannon index and $p_i$ is the proportion of cropland covered by crop $i$ out of the total cropland for all $n$ crops in the pixel.

To account for potential effects of agricultural management and expansion at a larger scale than the raster cell, we used percentage of natural habitat, as well as yield data from EarthStat and MapSPAM, to calculate an average percentage of natural habitat and yield in a buffer of one raster cell (~10-km radius) around the focal cell where the PREDICTS site was located. Because the average yield/percentage of natural habitat in the buffer was highly correlated with the value in the focal raster cell, we used the difference between the buffer and the focal cell value (buffer–focal difference) in the statistical models.

To account for the duration of substantial human landscape modification[59], we used the History database of the Global Environment (HYDE) database[111,112] to calculate the number of years since 30% of a grid cell (0.5° × 0.5° resolution) is estimated to have become covered by human land uses. Finally, we added climate variables in our models because climate is an important determinant of biodiversity. We extracted climate information based on the WorldClim estimates of historical annual mean temperature and annual precipitation for the period 1970–2000[113].

We log$_e$-transformed variables with right-skewed distributions: annual precipitation, MapSPAM yield, MapSPAM subsistence for all crops, EPIC-BOKU subsistence for maize and wheat, and EarthStat yield for rice. We added a value of 0.01 (less than 1% of the highest value) to deal with 0 values for MapSPAM subsistence yields for all crops, and for EPIC-BOKU subsistence-yield estimates for maize. Variables with left-skewed distributions were transformed with the formula:

$$V_t = \log_e(C - V), \qquad (2)$$

where $V_t$ is the transformed variable and $C$ is a constant used to avoid negative values, and determined by adding a small number to the highest value of the respective variable and rounding it up. Variables with a left-tail distribution were duration of substantial human landscape modification for all crops (with $C = 2{,}005$, obtained by rounding up the sum of the maximum value 1,981.6 and 20) and annual mean temperature for rice (with $C = 30$, obtained by rounding up the sum of the maximum value 28.5° and 1°). This transformation flips the order of the values of these two variables (that is, the highest values become the lowest) and the direction of their statistical effects. We then scaled all continuous variables to have a mean of 0 and a standard deviation of 1.

## Data processing for statistical analysis

All spatial estimates were transformed to the Behrman equal-area projection and resampled to the EarthStat resolution and extent in R with the projectRaster and resample functions, respectively, from the raster package, version 3.6–23[114]. We used bilinear interpolation for spatial averaging. We then extracted the values within spatial layers at the locations of the PREDICTS sites based on geographical coordinates. For the yield–biodiversity models for each crop, we selected PREDICTS sites that fell within pixels that contained cultivated areas with the respective crop according to EarthStat estimates. The PREDICTS cropland classification does not differentiate according to the crop cultivated at the precise sampling location. Therefore, the biodiversity data were sampled in landscapes used to grow the focal crops (according to EarthStat estimates), even if the actual sample may have been taken in a different crop. In doing so, we assume that intensification levels tend to be similar across different crops within an agricultural landscape.

For the MapSPAM yield, we chose either the 2000, 2005 or 2010 estimates based on which of the three years was closest to the year of the midpoint of the collection dates for each PREDICTS site. If a midpoint year was equally close to two MapSPAM years, we chose the earlier year because that would be more likely to represent the agricultural management impacting biodiversity by virtue of preceding in time.

## Statistical analysis

We modelled the three biodiversity metrics—species richness, abundance and RCAR—using mixed-effects models with random effects for study identity and spatial block nested within study, to account for non-independence of sites. The random effect for study accounted for differences in sampling methods, sampling effort, focal taxonomic group and broad geographic regions among the different studies in PREDICTS. The random effect for spatial block accounted for the spatial structuring of sampling within a study. To account for overdispersion, we included a random effect for site in the models for species richness[115]. To reduce the right skew in the abundance data due to measurements across many taxonomic groups and with a variety of methods, we scaled total abundance by dividing values by the maximum abundance within each study[55]. We then log$_e$-transformed the rescaled values to further reduce the skew of the distribution adding 0.01 to deal with 0 values.

For the yield–biodiversity models, each data point/site was weighted in the statistical analysis by the area of the respective crop as a fraction of all crops grown in the pixel to account for the differences in crop composition of agricultural landscapes according to the formula:

$$RA_C = \frac{A_C}{\sum_{i=1}^{175} A_i} \qquad (3)$$

where $RA_C$ is relative area for crop $C$, where $C$ is maize, soybean, wheat or rice, $A_C$ is the harvested area in hectares for crop $C$ and $A_i$ is the harvested area for each of the 175 crops available in the EarthStat dataset. We rescaled the area fractions such that the average weight is equal to 1, to avoid an apparent reduction in total sample size, which would have an adverse influence on some statistical properties:

$$\text{Rescaled\_RA}_C = N_{\text{Sites}} \frac{RA_C}{\sum_{j=1}^{N_{\text{Sites}}} RA_j} \qquad (4)$$

where $RA_j$ is the relative area for a given crop in each site $j$ of the total $N_{\text{Sites}}$ where that crop is cultivated.

The default set of initial variables for the land-conversion models was:

BD = LU–LS categories + Duration of human landscape modification

 +Mean annual temperature + Annual precipitation

 +Geographic region + LU–LS categories

 ×Duration of human landscape modification

 +LU–LS categories × Geographic region

$$\qquad (5)$$

where BD is one of the three biodiversity metrics and LU–LS categories represent the combined land-use–landscape categories. We included interaction terms between the combined land-use–landscape variable, and geographic region and duration of substantial human landscape modification to account for the clustering of natural and modified landscapes according to geographical patterns and historical human land use[111].

The default set of initial variables for the yield–biodiversity models was:

$$BD = Yield + \% \ natural \ habitat + Land \ use$$

$$+ Geographic \ region + Subsistence \ yield + Crop \ diversity$$

$$+ Yield \ buffer–focal \ difference$$

$$+ \% \ natural \ habitat \ buffer–focal \ difference$$

$$+ Duration \ of \ human \ landscape \ modification$$

$$+ Mean \ annual \ temperature + Annual \ precipitation \qquad (6)$$

$$+ Yield \times Subsistence \ yield + Yield \times Biome \ type$$

$$+ Yield \times \% \ natural \ habitat + Yield \times Land \ use$$

$$+ \% \ natural \ habitat \times Subsistence \ yield$$

$$+ \% \ natural \ habitat \times Biome \ type$$

$$+ \% \ natural \ habitat \times Land \ use$$

where Yield is the landscape yield for each of the four crops, % natural habitat represents the percentage of natural habitat in the landscape, and Yield buffer–focal difference and % natural habitat buffer–focal difference represent the differences between the one-cell buffer average value and the focal cell value for yield and percentage of natural habitat, respectively. All other variable names are self-explanatory. We considered yield and percentage of natural vegetation interactions with each other and with land use, geographic region and subsistence yield because research has suggested that these interactions impact biodiversity[17,55].

In cases of strongly correlated variables (correlation above 0.6), we removed one of them from the initial, full model by prioritizing yield, percentage of natural habitat and subsistence yield measures (in this order) to be kept for modelling (Supplementary Table 6). For the soy RCAR model, owing to model convergence issues we removed the following interactions: % natural habitat × subsistence yield, % natural habitat × biome type, % natural habitat × land use, yield × subsistence yield and yield × land use. We removed these interactions because they either have less support in the literature or they are less relevant for the focus of our study. Annual mean temperature was also removed as a main effect from the initial model structure. The convergence issues were probably triggered by a reduction in data points (1,482 RCAR data points compared with 2,404 for species richness).

For both yield–biodiversity and land-conversion models, we modelled species richness using generalized linear mixed-effects models with a Poisson distribution, while for $\log_e$-transformed abundance and $\log_{10}$-transformed RCAR we used linear mixed-effects models. We performed backward stepwise selection based on likelihood-ratio tests with a $P$-value threshold of 0.05 to select the fixed-effects structure for each model (Supplementary Table 7). As an alternative way to deal with overdispersion, we reran the species richness models substituting the Poisson distribution with a negative binomial distribution and dropping the random effect for site (Supplementary Table 20). To test the robustness of our results, we also ran the final model structures using a Bayesian modelling framework (Supplementary Table 21) with uninformative priors, 4 Markov chains and 5,000 iterations with 2,500 warmup iterations. We considered that the models converged if the Rhat convergence diagnostic was ≤1.01 and rank-normalized effective sample size was ≥400, which represents 100 times the number of chains[116]. For the species richness models, we had to increase the number of iterations to 6,000 to achieve convergence.

We conducted all statistical analyses and projections of biodiversity impacts of yield increases in R version 3.6.3[117]. For the mixed-effects modelling, we used the glmmTMB package, version 1.1.5[118]. For model selection, we adapted several functions from the packages associated with the PREDICTS data[119]. For checks of the models assuming a Poisson distribution we used DHARMa package[120], and for the Bayesian models we used the brms package[121].

## Projections of the impact of closing yield gaps

We used the best EarthStat–EPIC-BOKU models to project biodiversity change when closing yield gaps for the global areas occupied by the four crops in the year 2000. Yield gaps where defined by ref. 6 as the difference between attainable yields, which are the area-weighted 95th percentiles of observed yields within zones of similar annual precipitation and growing-degree days, and yields given by the Earth-Stat estimates for the year 2000 that are below attainable yield values for the area. We first projected biodiversity levels in cropland and primary vegetation for yield levels in the year 2000. We then used the same models to project biodiversity levels in cropland and primary vegetation at yields equal to those necessary to close yield gaps. We calculated an aggregated estimated biodiversity value across all four crops for each pixel, for each metric at each of the two yield levels, using the following formula:

$$BD = \sum_{i}^{n=4} p_i (BD_{Cropland} p_{Cropland} + BD_{PV} p_{PV}) \qquad (7)$$

where BD is the biodiversity metric in the respective pixel, $p_i$ is the proportion of crop $i$ in the total cropland of the respective pixel, $BD_{Cropland}$ and $BD_{PV}$ are values of the biodiversity metric estimated by the yield–biodiversity model for cropland and primary vegetation, respectively, and $p_{Cropland}$ and $p_{PV}$ are the proportion of each land-use type of the total area of the pixel. We calculated the percentage change in biodiversity relative to the biodiversity corresponding to the year 2000 yields by subtracting in each pixel the biodiversity metric at gap-closing yields from the biodiversity metric corresponding to year 2000 yield and dividing the difference by the biodiversity metric corresponding to year 2000 yields. The biodiversity contribution of the other land uses and crops in the pixel cannot be estimated by our models and, therefore, were not included in the calculation.

## Comparison of intensification and expansion impacts

We used the EarthStat–EPIC-BOKU models to project biodiversity metrics for expansion and intensification scenarios that would result in identical increases in total production in each pixel for each crop, including raster cells without a yield gap[6]. Although the available area for farmland expansion in each pixel might be less suitable than existing farmland, requiring more intense management to obtain similar yields, we are not able to consider such variation in our projections. Therefore, we made the assumption that the areas to be converted to farmland are of the same average suitability as existing local farmland.

We aggregated each biodiversity metric in each pixel for each crop according to the formula:

$$BD = BD_{Cropland} p_{Cropland} + BD_{PV} p_{PV} \qquad (8)$$

where BD is the biodiversity metric in the respective pixel, $BD_{Cropland}$ and $BD_{PV}$ are the model-estimated values of the biodiversity metric for cropland and primary vegetation, respectively, and $p_{Cropland}$ and $p_{PV}$ are the proportion of each land-use type of the total area of the pixel.

For the expansion scenario, we kept the yield equal to the year 2000 yield but increased the area cultivated with the respective crop by 1% at the cost of primary vegetation in the pixel. This meant also a decrease by the equivalent area in the total percentage of natural habitat. For the intensification scenario, the only variable that changed was the yield for each crop, which increased by 1% for the proportion of the grid cell equal to the proportion of the respective crop in the cropland area. The rest of the grid cell was modelled as having the same yield level as in 2000. We then calculated a percentage change in biodiversity by extracting the biodiversity calculated for the intensification scenario

from the biodiversity calculated for the expansion scenario and dividing the difference by the biodiversity metric corresponding to year 2000 yields and landscape composition.

We tested the robustness of our results to the choice of production increase by repeating the analysis for an increase in production of 10% and of 1% of local yield gap. We also tested the robustness of results to three different intensification patterns resulting in a total production increase of 1%: a 2% yield increase on 50% of the cropland area, a 10% yield increase on 10% of the cropland area, and a 10% yield increase on 91.8% of the cropland area and 'restoration' to primary vegetation of the remaining 8.2% of cropland area.

### Reporting summary

Further information on research design is available in the Nature Portfolio Reporting Summary linked to this article.

## Data availability

The PREDICTS database used for this study is available from https://data.nhm.ac.uk/dataset/the-2016-release-of-the-predicts-database. PREDICTS site-level biodiversity data with estimates of community-average range size are available via figshare at https://doi.org/10.6084/m9.figshare.7262732.v1 (ref. 122). The EarthStat data are available from http://www.earthstat.org/. The MapSPAM data are available from https://mapspam.info/. The EPIC-BOKU subsistence yield data are available via figshare at https://doi.org/10.6084/m9.figshare.25780953.v1 (ref. 123). The land-use data from ref. 98, on which we based our calculations of the percentage of natural habitat, can be downloaded from https://doi.org/10.4225/08/56DCD9249B224. The HYDE database[112] underlying the duration of substantial human modification data can be downloaded from https://landuse.sites.uu.nl/datasets/. The climate variables can be downloaded from https://www.worldclim.org/data/index.html. Source data for Fig. 4 are available via figshare at https://doi.org/10.6084/m9.figshare.28592318.v1 (ref. 124). Source data for Fig. 5 are available via figshare at https://doi.org/10.6084/m9.figshare.28592387.v1 (ref. 125). The datasets for running the statistical analyses are available via figshare at https://doi.org/10.6084/m9.figshare.28592393.v1 (ref. 126). Source data are provided with this paper.

## Code availability

The code required to run the analyses presented here can be downloaded from https://github.com/SilviaCeausu/BiodivYield.

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

## Acknowledgements

This work was supported by the UK Global Challenges Research Fund Trade, Development and Environment Hub (grant number ES/S008160/1). T.N. was also supported by a Royal Society University Research Fellowship (grant number UF150526).

## Author contributions

S.C. and T.N. conceived the study. S.C. and T.N. designed the study with contributions from D.L. D.L. facilitated access to a portion of the data used in the analysis. S.C. performed the analysis. S.C. wrote the first draft. All authors reviewed and edited the subsequent versions of the paper.

## Competing interests

The authors declare no competing interests.

## Additional information

**Extended data** is available for this paper at https://doi.org/10.1038/s41559-025-02691-x.

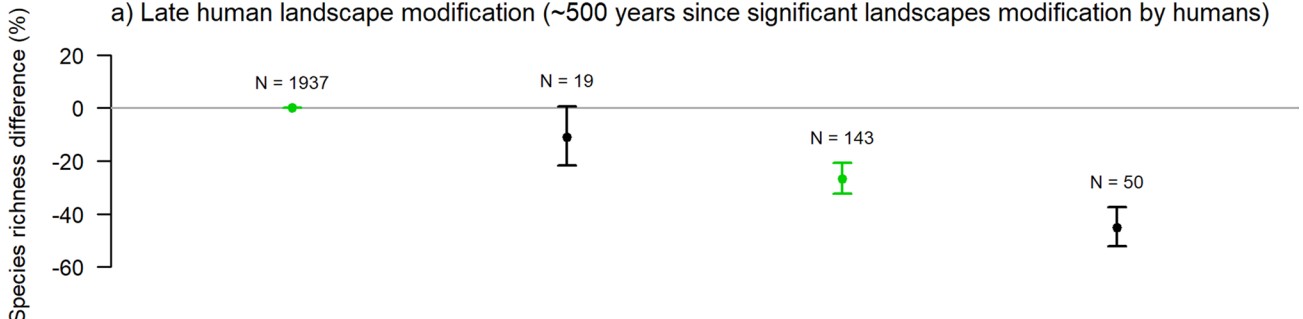

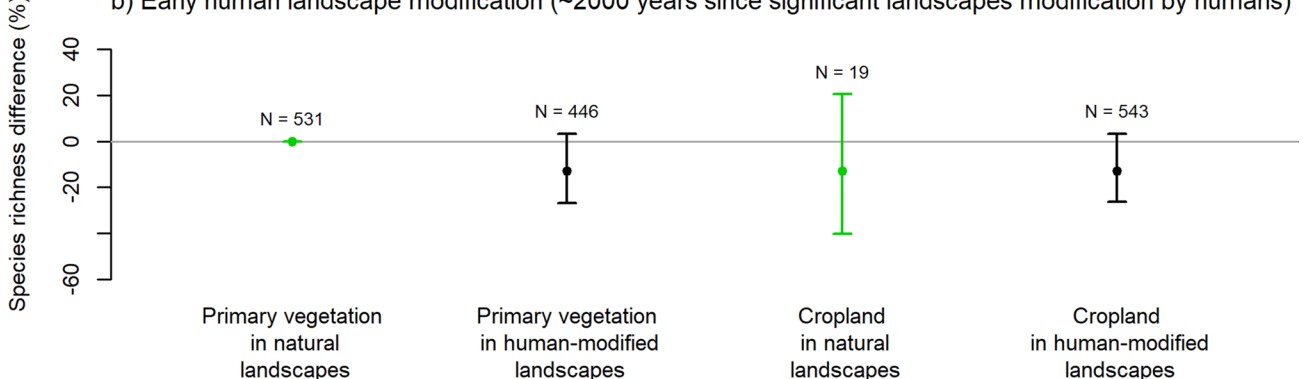

**Extended Data Fig. 1 | The modelled effect of land conversion on species richness in landscapes with different histories of substantial human landscape modification.** The modelled effect on species richness (SR) of land conversion at local and landscape scale for areas that experienced late substantial human landscape modification (~500 years ago) (**a**) and areas that experienced early substantial human landscape modification (~2000 years ago) (**b**) in primary vegetation and cropland, in natural landscapes (>70% natural habitat, green dots and error bars) and in human-modified landscapes (<30% natural habitat, black dots and error bars). Models are fitted with PREDICTS data sampled in primary vegetation and cropland. Median estimated values (points), and 2.5th and 97.5th percentiles (error bars) were calculated by sampling the fixed effects of the conversion models 1,000 times based on the variance-covariance matrix. The N values represent the number of sites for each class of the combined land use-landscape variable that were sampled in landscapes with at most 500 years (a) and at least 1950 years (b) since substantial human landscape modification, respectively.

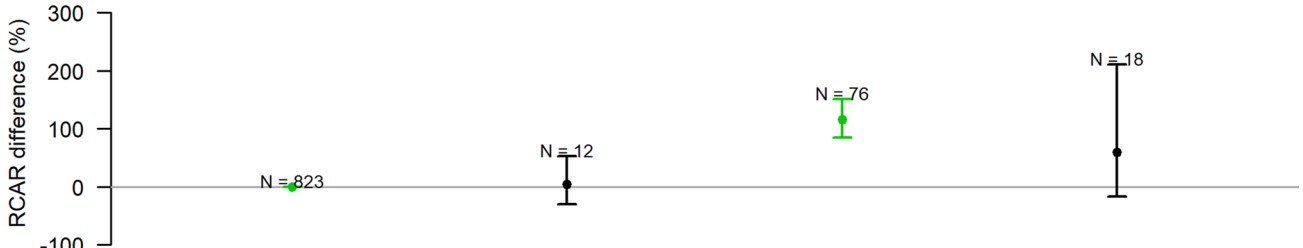

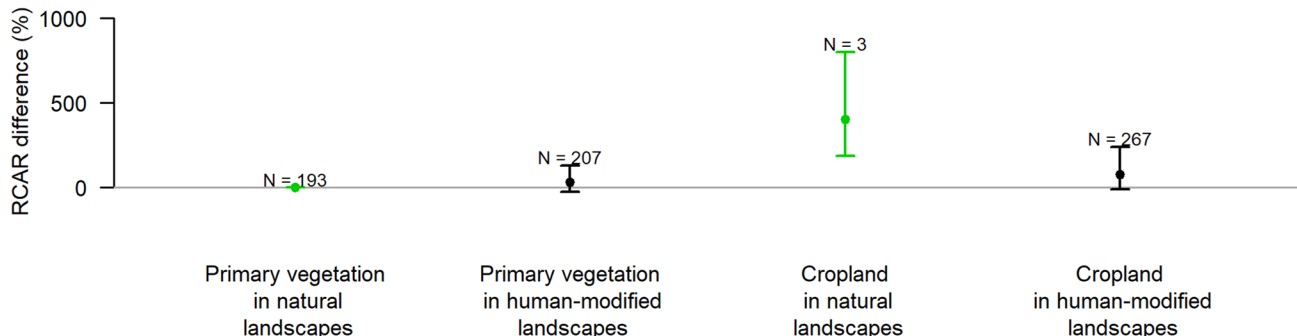

**Extended Data Fig. 2 | The modelled effect of land conversion on RCAR in landscapes with different histories of substantial human landscape modification.** The modelled effect on relative abundance-weighted average range-size (RCAR) of land conversion at local and landscape scale for areas that experienced late substantial human landscape modification (~500 years ago) (**a**) and areas that experienced early substantial human landscape modification (~2000 years ago) (**b**) in primary vegetation and cropland, in natural landscapes (>70% natural habitat, green dots and error bars) and in human-modified landscapes (<30% natural habitat, black dots and error bars). Models are fitted with PREDICTS data sampled in primary vegetation and cropland. Median estimated values (points), and 2.5th and 97.5th percentiles (error bars) were calculated by sampling the fixed effects of the conversion models 1,000 times based on the variance-covariance matrix. The N values represent the number of sites for each class of the combined land use-landscape variable that were sampled in landscapes with at most 500 years (a) and at least 1950 years (b) since substantial human landscape modification, respectively.

# Reporting Summary

## Statistics

For all statistical analyses, confirm that the following items are present in the figure legend, table legend, main text, or Methods section.

| n/a | Confirmed | |
|---|---|---|
| ☐ | ☒ | The exact sample size (*n*) for each experimental group/condition, given as a discrete number and unit of measurement |
| ☐ | ☒ | A statement on whether measurements were taken from distinct samples or whether the same sample was measured repeatedly |
| ☐ | ☒ | The statistical test(s) used AND whether they are one- or two-sided<br>*Only common tests should be described solely by name; describe more complex techniques in the Methods section.* |
| ☐ | ☒ | A description of all covariates tested |
| ☐ | ☒ | A description of any assumptions or corrections, such as tests of normality and adjustment for multiple comparisons |
| ☐ | ☒ | A full description of the statistical parameters including central tendency (e.g. means) or other basic estimates (e.g. regression coefficient) AND variation (e.g. standard deviation) or associated estimates of uncertainty (e.g. confidence intervals) |
| ☐ | ☒ | For null hypothesis testing, the test statistic (e.g. *F*, *t*, *r*) with confidence intervals, effect sizes, degrees of freedom and *P* value noted<br>*Give P values as exact values whenever suitable.* |
| ☐ | ☒ | For Bayesian analysis, information on the choice of priors and Markov chain Monte Carlo settings |
| ☐ | ☒ | For hierarchical and complex designs, identification of the appropriate level for tests and full reporting of outcomes |
| ☐ | ☒ | Estimates of effect sizes (e.g. Cohen's *d*, Pearson's *r*), indicating how they were calculated |

*Our web collection on statistics for biologists contains articles on many of the points above.*

## Software and code

Policy information about availability of computer code

| | |
|---|---|
| Data collection | This study uses existing datasets. No software was used to collect the data specifically for this study. |
| Data analysis | R 3.6.3 , Packages: glmmTMB version 1.1.7 , DHARMa 0.4.6, brms 2.21.0,  raster 3.6-23, StatisticalModels 0.1, predictsFunctions 1.0. The code required to run the analyses presented here can be downloaded from: https://github.com/SilviaCeausu/BiodivYield. |

For manuscripts utilizing custom algorithms or software that are central to the research but not yet described in published literature, software must be made available to editors and reviewers. We strongly encourage code deposition in a community repository (e.g. GitHub). See the Nature Portfolio guidelines for submitting code & software for further information.

## Data

Policy information about availability of data

All manuscripts must include a data availability statement. This statement should provide the following information, where applicable:
- Accession codes, unique identifiers, or web links for publicly available datasets
- A description of any restrictions on data availability
- For clinical datasets or third party data, please ensure that the statement adheres to our policy

The PREDICTS database used for this study is available from https://data.nhm.ac.uk/dataset/the-2016-release-of-the-predicts-database. PREDICTS site-level biodiversity data with estimates of community-average range size is available from https://figshare.com/articles/dataset/PREDICTS_site-level_biodiversity_data_with_estimates_of_community-average_range_size/7262732. The EarthStat data is available from http://www.earthstat.org/. The

# Research involving human participants, their data, or biological material

Policy information about studies with human participants or human data. See also policy information about sex, gender (identity/presentation), and sexual orientation and race, ethnicity and racism.

| | |
|---|---|
| Reporting on sex and gender | NA |
| Reporting on race, ethnicity, or other socially relevant groupings | NA |
| Population characteristics | NA |
| Recruitment | NA |
| Ethics oversight | NA |

Note that full information on the approval of the study protocol must also be provided in the manuscript.

# Field-specific reporting

Please select the one below that is the best fit for your research. If you are not sure, read the appropriate sections before making your selection.

☐ Life sciences    ☐ Behavioural & social sciences    ☒ Ecological, evolutionary & environmental sciences

For a reference copy of the document with all sections, see nature.com/documents/nr-reporting-summary-flat.pdf

# Ecological, evolutionary & environmental sciences study design

All studies must disclose on these points even when the disclosure is negative.

| | |
|---|---|
| Study description | This study provides a quantitative analysis of biodiversity effects to land conversion and yield increases, including closing yield gaps. The land conversion (independent of yield increases) analysis focused on cropland and primary vegetation in either largely unmodified landscapes or in highly modified landscapes. The yield increases analysis (accounting for amount of natural habitat) focuses on 4 crops: maize, soy, wheat and rice. We also use projections of the biodiversity effects models to compare farmland expansion and intensification in agricultural landscapes. <br> We use a subset of 1,318,867 cropland and primary vegetation records from 10,094 sites of the freely available PREDICTS database alongside publicly available yield and land-use data. These records were further selected based on landscape composition for the land conversion models (5328 sites), and to match the extent of maize (4862 sites), soy (2404 sites), wheat (3227 sites) and rice (2810 sites) cultivation extent according to the publicly available EarthStat data, which we used for yield and yield gap information. The sites in the biodiversity data are nested within spatial blocks, which are nested within studies. We build mixed effects models for three biodiversity metrics: sampled species richness, total sampled relative abundance, and relative abundance-weighted community-average range size (RCAR). The models included interactions between yield and natural habitat, and interactions of each of these two with subsistence yield, land use and biome. |
| Research sample | We used the 2016 release of the PREDICTS database, which contains 3,250,404 biodiversity records, mostly sampled from 2000 to 2012, from 666 published studies. Each study within the PREDICTS database contains data sampled with the same method across a gradient of land use or land-use intensity. The data in each study are grouped into one or more spatial blocks, each containing data from one or more sites. Each site is attributed one of 6 predominant land-use classes based on the information provided in the original papers, or by the authors of those papers. The subset of the data used in this study includes 18,853 species of which 3,994 |

are vertebrates, 6,693 are invertebrates, 7,269 are plants, 894 are fungi, and three are protists.
The PREDICTS database used for this study is available from https://data.nhm.ac.uk/dataset/the-2016-release-of-the-predicts-database. PREDICTS site-level biodiversity data with estimates of community-average range size is available from https://figshare.com/articles/dataset/PREDICTS_site-level_biodiversity_data_with_estimates_of_community-average_range_size/7262732. The EarthStat data is available from http://www.earthstat.org/. The land-use data based on which we calculated the percentage of natural habitat can be downloaded from http://doi.org/10.4225/08/56DCD9249B224.

| Sampling strategy | For the land conversion analysis, we used all PREDICTS cropland and primary vegetation sites that were in landscapes with either more than 70% natural habitat (unmodified landscape) or in landscape with less than 30% natural habitat (modified landscape). For the yield analysis, we used all sites that had relatively high quality information on yield based on EarthStat data for maize, soy, wheat and rice. |
| --- | --- |
| Data collection | All the datasets used for this analysis were either collected or modelled as described in the methods of this article. No new data were collected for this study. |
| Timing and spatial scale | Most biodiversity data were collected from 2000 to 2012. The frequency and periodicity differs for each study included in the PREDICTS database. EarthStat data represent yield, area and yield gap estimates for the year 2000, which are based on information for the years between 1997 and 2003. The land-use data that we used to estimate amount of natural habitat are estimates for the year 2005. Both the EarthStat and land-use data years are within the sampling timespan for the PREDICTS data. |
| Data exclusions | We only excluded data points that intersected with EarthStat yield estimates that were based on lower quality information as described in the methods section. This exclusion criteria was pre-established. |
| Reproducibility | All datasets and code are freely available. |
| Randomization | All data has been collected and published before this study. Therefore, randomization was not possible. Differences due to study design were accounted for by the use random effects. Covariates were controlled for by inclusion in the statistical models. |
| Blinding | Blinding was not possible for this analysis because all the data were collected before the start of this study. |

Did the study involve field work? ☐ Yes ☒ No

# Reporting for specific materials, systems and methods

We require information from authors about some types of materials, experimental systems and methods used in many studies. Here, indicate whether each material, system or method listed is relevant to your study. If you are not sure if a list item applies to your research, read the appropriate section before selecting a response.

## Materials & experimental systems

| n/a | Involved in the study |
| --- | --- |
| ☒ | Antibodies |
| ☒ | Eukaryotic cell lines |
| ☒ | Palaeontology and archaeology |
| ☒ | Animals and other organisms |
| ☒ | Clinical data |
| ☒ | Dual use research of concern |
| ☒ | Plants |

## Methods

| n/a | Involved in the study |
| --- | --- |
| ☒ | ChIP-seq |
| ☒ | Flow cytometry |
| ☒ | MRI-based neuroimaging |

## Plants

| Seed stocks | *Report on the source of all seed stocks or other plant material used. If applicable, state the seed stock centre and catalogue number. If plant specimens were collected from the field, describe the collection location, date and sampling procedures.* |
| --- | --- |
| Novel plant genotypes | *Describe the methods by which all novel plant genotypes were produced. This includes those generated by transgenic approaches, gene editing, chemical/radiation-based mutagenesis and hybridization. For transgenic lines, describe the transformation method, the number of independent lines analyzed and the generation upon which experiments were performed. For gene-edited lines, describe the editor used, the endogenous sequence targeted for editing, the targeting guide RNA sequence (if applicable) and how the editor was applied.* |
| Authentication | *Describe any authentication procedures for each seed stock used or novel genotype generated. Describe any experiments used to assess the effect of a mutation and, where applicable, how potential secondary effects (e.g. second site T-DNA insertions, mosiacism, off-target gene editing) were examined.* |

