## [Peer Review File · Nature Ecology & Evolution]

Geography and availability of natural habitat determine whether cropland intensification or expansion is more detrimental to biodiversity

Corresponding Author: Dr Silvia Ceausu

Version 0:

Decision Letter:

Dear Dr Ceausu,

Your manuscript entitled "Biome type and availability of natural vegetation dictate whether farmland intensification or expansion is worse for biodiversity" has now been seen by three reviewers. As you will see from the reports below, the reviewers find your work of interest but raise a number of concerns which will need to be addressed before we can offer publication in Nature Ecology & Evolution. Therefore, we would like to see a point-by-point response to the reviewers' comments and a revised manuscript.

* Please highlight all changes in the manuscript text file.

* If you have not done so already please begin to revise your manuscript so that it conforms to our Article format instructions at <http://www.nature.com/natecolevol/info/final-submission>. Refer also to any guidelines provided in this letter.

Link Redacted

We hope to receive your revised manuscript within 8 weeks. If you need a substantial extension, please let us know. We will be happy to consider your revision so long as nothing similar has been accepted for publication at Nature Ecology & Evolution or published elsewhere.

Nature Ecology & Evolution is committed to improving transparency in authorship. As part of our efforts in this direction, we are now requesting that all authors identified as 'corresponding author' on published papers create and link their Open Researcher and Contributor Identifier (ORCID) with their account on the Manuscript Tracking System (MTS), prior to acceptance. ORCID helps the scientific community achieve unambiguous attribution of all scholarly contributions. You can create and link your ORCID from the home page of the MTS by clicking on 'Modify my Springer Nature account'. For more information please visit www.springernature.com/orcid.

We look forward to seeing the revised manuscript. Thank you for the opportunity to review your work.

[redacted]

Reviewers' comments:

Reviewer #1 (Remarks to the Author):

This work provides an analysis of the impact on biodiversity of two strategies to increase agricultural output: farmland intensification and farmland expansion. The authors effectively delineate and explain the problem in the abstract and introduction, although there is a need for clear definitions of some terminology. For example, total "total abundance" is mentioned in lines 21 and 72-73, but it is not given an example of what it is how it measured (number of individuals, total biomass?; see also below the comment about lines 128-130, 18 and 118-119 below). The Methods subsections are somewhat tedious and lengthy. I understand the importance of these sections, and I commend the authors for providing a very detail information. However, as it stands, it disrupts the flow of the article; this could be improved by using tables and moving some of the more detailed aspects to the Supplementary Material.

My main question is:

- How much of the conclusions can be generalized, given that only the impact of four crop types were assessed? While the authors provide a thorough discussion on the limitations of their approach, I did not see how they address this specific concern.

Other questions:

- I am rather surprised by the simultaneous use of frequentist and Bayesian statistics. This seems excessive and suggests a lack of confidence in one or both approaches.

- In line 642, under the section "Biodiversity impact of closing yield gaps" and in figure 4, I found the opposite effects on total abundance (figure 4b) and RCAR in the tropics (figure 4c) rather striking. Additionally, the contrasting effects between species richness and abundance (figures 4a, 4b) and RCAR (figure 4c) for non-tropical regions are noteworthy. I do not think these aspects are sufficiently highlighted in the text.

Minor Details:

- Line 18 - "Closing yield gaps": If the authors want to reach a wide readership, including those unfamiliar with some of the terminology used in this field, they should provide a clear explanation of the meaning of several terms used, such as, "closing yield gaps." Although it may seem obvious, it is easy to misunderstand its meaning.
- Line 78: What is a "more ecological community"?
- Line 84: Why "maize, soybean, wheat, and rice" crops? The answer is on lines 185-186, but I would move it to the introduction.
- Line 115: It is mentioned "6 predominant land-use classes." Which ones?
- Lines 118-119: The definition of "primary vegetation" only appears here, but it has been mentioned before. I suggest moving it to the location where it first appears.
- Lines 128-130: It would be important to have some examples of measures of abundance.
- Line 270: The authors use "subsistence yields" as a proxy for "agricultural suitability." The rationale seems to be that the former does not require "inputs, irrigation, and most human management." If I understand correctly, this seems to be a rather narrow definition of "subsistence yields," because even subsistence yields often require some external inputs or irrigation. I suggest the authors provide a more thorough explanation for using "subsistence yields" as a proxy for "agricultural suitability."
- Line 377 - Section "Statistical Analysis": I would like to see the set of "equations" used. This would be a good way of summarizing the information that is, in fact, quite dense. These "equations" could simply be the instructions used in R to define the statistical models, or some schematic illustrations of how variables and parameters are related.
The authors say that "to deal with overdispersion, we reran the species richness models substituting the Poisson distribution with a negative binomial distribution and dropping the random effect for site." (my emphasis) Maybe this wasn't necessary given that the model is a mixed-effects model (a consequence that is more clearly understood under the Bayesian approach). In these models there is hierarchical structure where the mean of the Poisson is being modeled with some distribution. For example, when the mean of a Poisson distribution is modelled by a gamma distribution, then the resulting distribution is a negative binomial. In other words, when using a Poisson with a mixed-model/hierarchical model, the overdispersion is automatically being taken care of (e.g., the Wikipedia: https://en.wikipedia.org/wiki/Negative_binomial_distribution) or McElreath, R. (2020). Statistical rethinking: A Bayesian course with examples in R and Stan. Chapman and Hall/CRC). This may be somehow related to the authors' observation in lines 637-638.
- Line 591 – Section "Biodiversity impacts and land conversion": I suggest presenting the results of this section in a table.
- Line 692: Can you say that often "species with large global ranges" are generalists?
- Line 711: I found the paragraph starting on line 711 difficult to follow. The authors start by discussing "wheat landscapes," but then change to "maize and soybean landscapes" (line 719) and later to "rice landscapes" (line 725). However, the transition between these topics is not clear, and I initially assumed the authors were going to compare their impacts.
- Lines 726-728: I don't understand the conclusion: "This suggests that species richness and abundance increases with rice yield in landscapes with a high percentage of natural habitat is driven by narrow-ranged species."
- Lines 741-742: The value 99.7% seems excessively high.
- Line 785: There is a reference to straight lines (most likely correctly) and then a reference to Figures 4 and 5, but these do not depict straight lines. This is a little bit confusing because the relationship between the straight lines first mentioned and the results in the figures is not clear.

Figures:

- Figure 1a: Why do some "dots" have different color intensities (e.g., in South America)?

- Figure 1b: The relative size of the circles is too similar. Why are some points circumferences and others circles?
 - Figure 1c: I don't understand this plot. What is the meaning of the "axes"? What is the "grey square"?
 - Figure 2: I see no need to show the green dots for "Primary vegetation in natural landscapes." This is just a reference value that is implicit in the analyses. (The labels on the y-axis are not fully visible.)
- Supplementary Material: Supporting Figures:
- What is the meaning of the dashed red lines in figures S5-S12?
 - ATTENTION: Some of the figures have the same "number" (two figures S5, two figures S9, and two S14).

Reviewer #2 (Remarks to the Author):

In their manuscript titled "Biome type and availability of natural vegetation dictate whether farmland intensification or expansion is worse for biodiversity", the authors compare the impact of agricultural expansion and intensification scenarios on biodiversity, represented by metrics for species richness, total abundance and the relative community abundance-weighted average range size (RCAR). In order to compare the scenarios, the authors assume a production increase target of +1% at each pixel that is obtained either by intensification or expansion. The authors address an important and relevant research question at the global scale. The paper is well written and basically uses appropriate data and methods, that could somehow be clearer presented.

General comments:

While the biodiversity analysis is well performed and explained, my main general concern is about the conceptual framework and underlying normative assumptions of the study. For comparison of intensification and expansion, both scenarios aim at reaching the same production increase target of +1%. This seems arbitrary to me, and it is not explained why 1% is chosen. Further, the assumptions underlying the two land use scenarios are very normative and I am afraid that the assumptions also largely determine the results. The authors write that their chosen scenarios are not realistic projections for the future, but rather capturing an extreme case of possible strategies to increase agricultural production. Nevertheless, there is no theory behind, what is driving agricultural expansion/intensification and where this is gonna happen. Changing these assumptions could probably lead to very different results and conclusions, which affects the robustness of the approach. As an example, expansion is only allowed into primary vegetation and takes place at each pixel up to the 1% production increase target in the comparison scenario. The 1% production increase target applies to every pixel. Why e.g. not implementing a global production increase target? Yields are increased uniformly on each pixel by 1%, unless a yield increase by 1% in regions with low yields (high yield gaps) increases absolute yields only very marginally. Therefore, closing yield gaps by a certain percentage value would have been more intuitive. It would also be more realistic to assume that yield increases are stronger in regions with high yield gap, due to the fact that yield gaps are already almost reached (and even above in some regions) in high productive regions. Finally, you conclude that neither intensification nor expansion is consistently better for biodiversity in context of the land sparing land sharing debate. However, the intensification approach does not consider that less land is used for agriculture and could therefore be given back to natural ecosystems, because in your approach, you uniformly intensify all cropland by 1% within a pixel (at landscape scale), but also regionally or globally, which to my understanding is not land sparing, which is a compaction of used land that goes along with a renaturation of marginal lands, not only an intensification (compare e.g. Schneider et al. 2022, <https://doi.org/10.1371/journal.pone.0263063>). In case of expansion, more land is used and no land is taken out of production and restored to natural ecosystems, which is a major reason why land use expansion in average is worse for biodiversity, at least if you look at this at the landscape scale. Accordingly, linking your approach to the land sparing/land sharing debate is difficult, and therefore, in my opinion, your conclusion is biased in that regard. What you would have to do is to intensify in each pixel e.g. by 10%, then reducing the used land to reach the 1% production increase target and restore natural ecosystems on the freed up land. Would that be possible?

Data and Methods: It is not easy to follow the methodology at first reading due to the many different data sets. I had to read several sections twice. An overview is missing, which would help a lot to quickly understanding the used data and methods. Therefore, I suggest showing a schematic figure for both, the yield-biodiversity models and the land-conversion models, giving an overview on the applied data and how they were used.

The time period the authors target their scenario is approximately the year 2000 and not a future projection, considering climate change, etc., which would make the study more interesting.

Instead of using potential yield estimates by Mueller, that are based on observed yields within AEZs, you could use potential yields, simulated by crop models, which take spatial heterogeneity and future climate change impacts into account. They tend to be higher than observation based approaches, especially in developing countries.

In the discussion, I would additionally expect one sentence saying that also other ecosystem services rather than biodiversity must be considered in an integrative approach, such as carbon storage and water infiltration, etc.

I am wondering if you consider any kind of maximum expansion potential, which limits agricultural expansion mainly in bread basket regions, because available areas for expansion are very limited here.

While the study gives interesting new surprising findings and a new view on the effects of land intensification on agriculture, I am still not fully convinced that these findings are robust and not a result of the assumptions and the conceptual framework of the approach. This needs to be clarified. Therefore, I suggest major revisions of the manuscript.

Minor comments:

Title:

I suggest to use the term “cropland” instead of “farmland” in the title, because farmland could include pasture areas or rangeland which is not considered in your approach.

Abstract:

While you quantify the effect of intensification on biodiversity in the abstract by a median species loss of 11%, you just say that land conversion is associated with significant biodiversity loss”. I suggest to give quantitative results for both aspects for comparison, which would improve the significance and informative value of the abstract.

Figures:

Figure titles are used instead of figure subtitles, which confused me. Maps are not well illustrated. I’d suggest to use a global frame and cut at -60° South (showing all global maps without Antarctica).

Specific comments:

Ln 25: “yield increases also lead to” instead of “yield also lead to ...”

Ln 42: Which has already been done by some other studies, e.g. Zabel et al. (2019). <https://doi.org/10.1038/s41467-019-10775-z>

Ln 51: , whereas

Ln 66-70: This is an important aspect, referring to the discussion on sustainable intensification, which could also be mentioned here as a buzzword. Nevertheless, sustainable intensification is not considered in your approach, which I assume, would impact on the comparison between expansion and intensification quite drastically, also to the authors’ conclusions. That should also be added to the discussion. Also, I would be interested in an explanation, what kind of intensification this data driven statistical approach implies in different regions and how this could impact on biodiversity?

Ln 211-213: At least for rice, I assume that rainfed systems are also not used by subsistence farmers, but honestly I don’t know.

Ln 364-367: This is an important limitation of this study, because usually cash crops might be used at much higher intensity level than other crops. Also, I guess that there is a huge difference between crops. I was wondering if ‘cropping system’ is the right term here? Maybe, ‘across different crops’ might be more accurate?

Ln 452: blank before full stop.

Ln 476-478: This is an important assumption for understanding the whole study and I have been missing this information while reading the chapters before. I was always wondering, where expansion and intensification takes place and by how much you close yield gaps, etc. Now this is explained here. You expand and close yield gaps to reach a uniform production increase by 1% for each pixel and each crop. Maybe mention that already earlier, maybe end of introduction where you describe the general study framework.

Line 484: If you increase the area by 1%, you assume homogeneous yields within a pixel, which may not be correct. I guess this overestimates yields, because usually, the best land is already in use and cropland is expanded into less productive land, at least in human modified landscapes. Another issue might be that expanded land often is used at lower intensity, while you assume the same yield gap than for the rest of the area within the pixel, which may not be the case. These assumptions should be addressed and/or discussed.

Ln 488: In the previous part of the article, including the abstract, you mention closing the yield gap. Also Figure 4 is showing the effects of closing yield gaps (not saying by how much, I assume fully closing). However, increasing yields uniformly by 1% is s.th. else than closing yield gaps by 1%. You do not explicitly say that you close yield gaps by 1%, but at least reading the previous parts, it gives the impression and therefore, the text is somehow confusing in this context.

Another question relating to these very small intensification levels in regions with large yield gaps is the question if the statistical approach is sensitive enough to capture the impacts on biodiversity of such small absolute yield changes? Especially for subsistence farmers, inter-annual yield variabilities are much higher than 1% meaning that a yield increase of 1% cannot significantly be related to changes in management.

Ln 592-602: Not clear at first reading: Does this refer to a land conversion of all primary vegetation or to the +1% production increase scenario?

Appendix Figure S23. Source missing.

Reviewer #3 (Remarks to the Author):

Opinion on the paper by Silvia Ceaușu, David Leclère, Tim Newbold: Biome type and availability of natural vegetation dictate whether farmland intensification or expansion is worse for biodiversity

Description and overall assessment

This global study examines the biodiversity effects of land conversion, and of two major strategies, namely increasing yield on the same land area, or expanding farmed area without intensification. These questions have been explored before. For instance, in a recent study by Jaureguiberry et al (2022, Science Adv), they found that land/sea conversion is the primary cause of biodiversity loss. The two major strategies, known as the "land sharing / land sparing dilemma," have also been the subject of numerous articles, many of which have concluded that there is no easy answer because it depends on the context. Therefore, I have some doubts regarding the novelty of these results. Nevertheless, it utilized quantitative ecology data from a diverse array of taxa, indicators and locations, thus, being probably the most comprehensive study, utilizing several global database/estimates.

Major comments

This study is extensive, with Figure 3 alone providing a broad variety of results to analyse, like why are there many relationships (like for soy and RCAR), while none in others (wheat and RCAR). The article analysed biome, crop, biodiversity indicators, landscape context, land use, and their interactions. However, only a limited number of results were emphasized and described in the paper.

I find it difficult to fully endorse the prioritization of yield over factors such as fertiliser consumption, particularly when the focus is solely on biodiversity benefits and socio-economic aspects are not taken into account in the study. Productive land can achieve high yields without relying heavily on agrochemicals, but low quality areas may experience low yields despite using large amounts of fertilizers. Consequently, in the second scenario, the yield is low, but the negative impact on biodiversity is greater.

Biotic homogenisation is a measure of community composition, restricted to spatial context. Note, however, that this not characterise the species in the community, as range sizes are influenced by human activities, e.g. via extirpation of suitable habitats.

Why 1% increase in total production was targeted? 5% or 10% increase would be also interesting. As we now that c 800 million people are starving and we are 8 billions, a 10% would be realistic. On the other hand, however, if food system would be effective (no waste, no overconsumption /c. 900 million live with obesity/, etc.), what biodiversity gain would be if 1%, 5% or 10% decrease in yield will be possible?

I enjoyed the presentation of study limitations (from line 773), as it listed several of my concerns I noted while reading the manuscript, like missing of local scale, data/database quality, straight line across the maps. So this is a valuable part of the paper, clarifying its uncertainties. But these exist. In addition, the models not fully fit assumptions. Thus, results need some caution when interpreting.

Further comments:

What is closing the yield gap? Define.

Line 42: From this point, the introduction is on land sharing/sparing. However, these terms are ignored from the abstract, making a little bit of mismatch between these sections.

Line 56-57. „Neither species persistence or community metrics response to yield increases were assessed globally” it is true but, Martin et al (2019, Ecol Letters) published a large study on yield and landscape at the European scale.

Line 97-98: why 30 and 70% are the shares?

Line 123: where are Table S1-2-3? Not elegant to start with Table S4.

Figure 1. Looking at the map, I have the impression that the distribution of sites is aggregated, that is similar sites are clustered spatially. E.g. natural localities in natural landscapes are in SE Asia and Oceania, while Europe is the opposite. Thus, a conclusion that biodiversity decline is due to human ecosystem modifications can also be a spatial effect, that is the "treatments" are in different biogeographic regions. How to separate geography and human influence variables? There are also pale blue circles (in Chile and Argentina, Japan, NE America). Is this a typo, or has some meaning? 1/c figure. I cannot understand it....

Fig.2. Increase the distance between a) and b), otherwise there is overlap in Y axis legend.

Fig.2. On a figure with values and error bars, the reader will look for significances, which are missing from this figure, thus, it is not easy to understand where are the differences or similarities?

Fig.3. It has an extreme amount of information, really need time to digest it. Interesting to see that two graphs are empty, while others have a lot relationships plotted.

Also interesting to see that abundance and RCAR show similar trends in maize and soy, indicating more individuals in more homogenised communities  a potential threat of invasions parallel with yield increase?

Fig.3. line 538. This was 70% versus 30% above (line 97-98). Why different numbers?

In figure 3 several relationships show only one curve, that with the most data points. It ease illustration and understanding on a complex figure, but are the other relationships presented somewhere? Did not the reader miss some important results?

Line 604: I would not say most often, as from the four crops it is true for two (missing in soy, and contrasting in rice).

Line 606: typo: space is missing.

Line 645: Yes, but the blue colour (increase of richness) is rather clearly dominates tropical areas, like S Amazonia, and SE Asia. So the truth that increase in yield in the tropical areas has both benefits and costs on richness, while mostly costs in the temperate zone.

L 679-680: The role of landscape context was multiple times published see e.g. Batáry et al. (2011, Proc R Soc).

L763: Others concluded that multifunctional landscapes are needed, a combination of sharing and sparing lands (Grass et al 2021, Adv Ecol Res).

Supplementary tables: there was probably some converting problem, a few letters were misplaced by „?“.

Explain what is SR as this acronym was not used before.

RCAR 55km is also new name (it was RCAR before).

Table S7, first column, correct column width to have „abundance” in one line.

Supplementary figs: Fig S23: what is the dimension of the scale? %?

Version 1:

Decision Letter:

29th January 2025

Dear Dr. Ceausu,

Thank you for your patience during the second round of review of your manuscript "Geography and availability of natural habitat determine whether cropland intensification or expansion is more detrimental to biodiversity" (NATECOLEVOL-24041197A). Based on the feedback from the reviewers, whose reports will find below, we will be happy in principle to publish it in Nature Ecology & Evolution, pending minor revisions to satisfy the reviewers' final requests and to comply with our editorial and formatting guidelines.

[redacted]

Reviewer #1 (Remarks to the Author):

I congratulate the authors by the changes.

Two small details:

First, I think the legend of Figure 1 is missing as well as figure 2.

Line 503, I think there is a comma missing between "cropland" and "pasture" (otherwise it seems that "cropland pasture" is a class).

Reviewer #2 (Remarks to the Author):

The authors provided a detailed reply and extensive revisions on all points. Most importantly they have addressed my scepticism regarding the conceptual framework of the approach and that normative assumptions might influence the results. Therefore, additional analyses were carried out to demonstrate the robustness of the results. Other aspects, that haven't been addressed before, are now added to the discussion. I am a little bit surprised that some of the explanations given to my comments and figures are not included to the discussion, or added to the supplementary, since I think they would have been interesting for other readers as well. The paper structure improved a lot, especially the revisions at the end of the introduction help a lot understanding the study.

In Line 344, the authors write that the results of their study are consistent with previous studies on local land-use change and landscape composition effects. However, there is also a bunch of studies that come to different conclusions, maybe at different scale and also by using different approaches. Nevertheless, I think it is important and also interesting to place the results of this study in the context of these studies (see below) and discuss differences and implications. You could refer to these studies: Folberth et al. (2020) (<https://doi.org/10.1038/s41893-020-0505-x>), Schneider et al. (2022) (<https://doi.org/10.1371/journal.pone.0263063>), and Beyer et al. (2022) (<https://doi.org/10.1038/s43247-022-00360-6>) e.g. found

that global cropland could almost be halved within planetary boundaries by sustainable intensification. On the other side, Schneider et al. (2024) (<https://doi.org/10.1038/s41893-024-01410-x>) e.g. found that cropland expansion resulted in very high trade-offs regarding biodiversity and GHG emissions however using a very different approach. Foley et al. (<https://doi.org/10.1038/nature10452>) already showed in 2011 that halting agricultural expansion and closing yield gaps instead should be a priority for future land-use planning.

Other comments:

Line 42: The following paper calculates where land-use expansion pressure could increase for a future scenario: Schneider et al. (2024) <https://doi.org/10.1038/s41893-024-01410-x>. This could be cited here and is maybe also interesting for a follow-up study.

Line 350-357: Primary vegetation in modified landscapes may hold threatened species. This should be discussed, since none of your metrics is considering this explicitly.

Reviewer #3 (Remarks to the Author):

I appreciate the hard work of the authors to revise the paper. I have read the rebuttal letter and the revised version of the paper. My major concern was about the novelty of the results, which is now much clearer from the replies. I still feel it should be more clearly mentioned in the paper, but that may be too pushy, at least in the abstract. Regarding the application of yield as an indicator and not agrochemical use, my concerns are still valid, but I accept that the authors found yield more fitting to their study.

I still think homogenisation is not the best measure of community composition, as it measures only one character, the range size of species. I acknowledge, however, that this is an attempt to measure composition, which is usually neglected as an indicator, although usually more sensitive than richness or abundance.

On the 30% - 70% share Andr en's seminal paper is cited. That paper has limited relevance as it goes for birds and mammals, and on a limited range of habitats and regions. I may add the concept of Tscharntke on simple and complex landscapes. There are more or less similar percentages, but linked to agricultural landscapes.

Version 2:

Decision Letter:

25th March 2025

Dear Dr Ceausu,

We are pleased to inform you that your Article entitled "Geography and availability of natural habitat determine whether cropland intensification or expansion is more detrimental to biodiversity", has now been accepted for publication in Nature Ecology & Evolution.

Over the next few weeks, your paper will be copyedited to ensure that it conforms to Nature Ecology and Evolution style. Once your paper is typeset, you will receive an email with a link to choose the appropriate publishing options for your paper and our Author Services team will be in touch regarding any additional information that may be required.

Due to the importance of these deadlines, we ask you please us know now whether you will be difficult to contact over the next month. If this is the case, we ask you provide us with the contact information (email, phone and fax) of someone who will be able to check the proofs on your behalf, and who will be available to address any last-minute problems. Once your paper has been scheduled for online publication, the Nature press office will be in touch to confirm the details.

Acceptance of your manuscript is conditional on all authors' agreement with our publication policies (see www.nature.com/authors/policies/index.html). In particular your manuscript must not be published elsewhere and there must be no announcement of the work to any media outlet until the publication date (the day on which it is uploaded onto our web site).

Authors may need to take specific actions to achieve [compliance](https://www.springernature.com/gp/open-research/funding/policy-compliance-faqs) with funder and institutional open access mandates. If your research is supported by a funder that requires immediate open access (e.g. according to [Plan S principles](https://www.springernature.com/gp/open-research/plan-s-compliance)) then you should select the gold OA route, and we will direct you to the compliant route where possible. For authors selecting the subscription publication route, the journal's standard licensing terms will need to be accepted, including [self-archiving and license to publish](https://www.nature.com/nature-portfolio/editorial-policies/self-archiving-and-license-to-publish). Those licensing terms will supersede any other terms that the author or any third party may assert apply to any version of the manuscript.

If you have any questions about our publishing options, costs, Open Access requirements, or our legal forms, please contact

ASJournals@springernature.com

We welcome the submission of potential cover material (including a short caption of around 40 words) related to your manuscript; suggestions should be sent to Nature Ecology & Evolution as electronic files (the image should be 300 dpi at 210 x 297 mm in either TIFF or JPEG format). Please note that such pictures should be selected more for their aesthetic appeal than for their scientific content, and that colour images work better than black and white or grayscale images. Please do not try to design a cover with the Nature Ecology & Evolution logo etc., and please do not submit composites of images related to your work. I am sure you will understand that we cannot make any promise as to whether any of your suggestions might be selected for the cover of the journal.

You can generate the link yourself when you receive your article DOI by entering it here: <http://authors.springernature.com/share>.

[redacted]

P.S. Click on the following link if you would like to recommend Nature Ecology & Evolution to your librarian <http://www.nature.com/subscriptions/recommend.html#forms>

** Visit the Springer Nature Editorial and Publishing website at http://editorial-jobs.springernature.com?utm_source=ejP_NEcoE_email&utm_medium=ejP_NEcoE_email&utm_campaign=ejp_NEcoE for more information about our career opportunities. If you have any questions please click [here](mailto:editorial.publishing.jobs@springernature.com).

Reply to Reviewers' Comments

Geography and availability of natural habitat determine whether cropland intensification or expansion is more detrimental to biodiversity

We thank the three reviewers for their constructive comments.

In addition to addressing all comments as described below, we have also corrected two inconsistencies in our analysis that have led to minor changes in results:

- 1) *We realised that we had performed some additional, unnecessary processing steps when calculating percentage of natural vegetation based on land use data from Hoskins et al (2016)¹.*

Initially, to calculate percentage of natural vegetation, we projected primary vegetation and secondary vegetation cover estimates, onto a Behrmann equal-area projection. We then summed up these two proportional covers and calculated an average percentage of natural vegetation within a 5-km × 5-km grid, which is not the resolution that we used for our analysis. So, we then had to resampled these data again to the grid of the EarthStat data. For the rest of the analysis, we had projected the proportional cover estimates onto a Behrmann equal-area projection and then resampled them directly to the grid of the EarthStat data.

In the updated analysis, we calculated percentage natural vegetation directly at the resolution of the EarthStat data, as described on L 590 - 596.

- 2) *In the intensification-expansion comparison (L810-849), we realised that we were overestimating the impacts of intensification compared to expansion because the biodiversity change was estimated for yield changes on the entire cultivated area. In comparison, we estimated the expansion impacts for a cropland expansion that only took into account crop-specific cultivated area.*

We have now corrected this inconsistency by estimating biodiversity change for yield increases only on cropland cultivated with the respective crop as described on L837-840. This correction leads to larger proportions of cropland on which intensification is better for biodiversity than expansion. However, our overall conclusion remains the same because there are still sizable areas where our models predict higher biodiversity costs of expansion compared to intensification.

Finally, we have replaced everywhere in our draft "biome" with geographic region, which is a better description of the geographic classification used here.

Reviewers' comments:

Reviewer #1 (Remarks to the Author):

This work provides an analysis of the impact on biodiversity of two strategies to increase agricultural output: farmland intensification and farmland expansion. The authors effectively delineate and explain the problem in the abstract and introduction, although there is a need

for clear definitions of some terminology. For example, total “total abundance” is mentioned in lines 21 and 72-73, but it is not given an example of what it is how it measured (number of individuals, total biomass?; see also below the comment about lines 128-130, 18 and 118-119 below).

We thank the reviewer for their constructive comments. We have added information of how abundance was measured and percentage of sites with abundance information for each type of measurement (lines 516-518). We address the rest of the comments below.

The Methods subsections are somewhat tedious and lengthy. I understand the importance of these sections, and I commend the authors for providing a very detail information. However, as it stands, it disrupts the flow of the article; this could be improved by using tables and moving some of the more detailed aspects to the Supplementary Material.

In the new version of the manuscript, we have moved the Methods section to the end of the article, as indicated by the format of Nature Ecology & Evolution. We have also reduced the length of the section and moved some of the details in a Supplementary Methods Note. We have also added a new figure that replaces the previous figure 1c and which summarises the data usage and sources for the different models.

My main question is:

- How much of the conclusions can be generalized, given that only the impact of four crop types were assessed? While the authors provide a thorough discussion on the limitations of their approach, I did not see how they address this specific concern.

We thank the reviewer for bringing up this important point that we have missed in the initial manuscript. We have now added a discussion of this limitation to the Discussion section (L457-465):

“Moreover, our results are relevant for the set of crops considered here, and it is uncertain whether they would generalise to other crops. For example, agroforestry or multiannual cultivation systems might have different requirements for increasing yields^{24,27} and these requirements may have different impacts on biodiversity than those presented here. [...] In addition, our yield data might not capture effectively the intensity levels of the management of other crops in the landscape. We reduced the impact of this limitation by weighting our data points in the statistical analysis by the proportion of the respective crop in the landscape.”

Other questions:

- I am rather surprised by the simultaneous use of frequentist and Bayesian statistics. This seems excessive and suggests a lack of confidence in one or both approaches.

We thank the reviewer for this comment. We used both approaches to test the robustness of our results to different model fitting methods: maximum likelihood estimation by integrating over random effects for the frequentist approach (the glmmTMB package uses the Laplace approximation), and Markov chain Monte Carlo algorithms that generate random samples from the parameter distributions for the Bayesian approach. This type of robustness check has been previously used to increase confidence in statistical results of complex statistical models^{2,3}.

- In line 642, under the section "Biodiversity impact of closing yield gaps" and in figure 4, I found the opposite effects on total abundance (figure 4b) and RCAR in the tropics (figure 4c) rather striking. Additionally, the contrasting effects between species richness and abundance (figures 4a, 4b) and RCAR (figure 4c) for non-tropical regions are noteworthy. I do not think these aspects are sufficiently highlighted in the text.

We thank the reviewer for pointing out this gap in our text. We have now added an additional paragraph to the section "Biodiversity impact of closing yield gaps" that highlights the differences in impacts across metrics and biome types (lines 301-306).

These contrasting effects of closing yield gaps are the result of opposing effects of yield as described in the "Biodiversity impacts of yield increases" subsection (lines 258-299). These effects are further highlighted and discussed in the Discussion section (lines 358-382).

Minor Details:

- Line 18 - "Closing yield gaps": If the authors want to reach a wide readership, including those unfamiliar with some of the terminology used in this field, they should provide a clear explanation of the meaning of several terms used, such as, "closing yield gaps." Although it may seem obvious, it is easy to misunderstand its meaning.

We thank reviewer for highlighting this gap in our text. We have now added an explanation of the meaning of closing yield gaps on lines 87 - 88.

- Line 78: What is a "more ecological community"?

The text "more ecological communities" was meant to indicate that the species referred in this sentence were part of a higher number of ecological communities. We changed "more ecological communities" to "a higher number of ecological communities" to avoid confusion (lines 77 - 78).

- Line 84: Why "maize, soybean, wheat, and rice" crops? The answer is on lines 185-186, but I would move it to the introduction.

As suggested by the reviewer, we moved the information on the importance of maize, soybean, wheat and rice in the Introduction – lines 83 – 84.

- Line 115: It is mentioned "6 predominant land-use classes." Which ones?

As suggested by the reviewer, we have now listed on lines 502-503 the land-use classes in which sites were classified in the PREDICTS database.

- Lines 118-119: The definition of "primary vegetation" only appears here, but it has been mentioned before. I suggest moving it to the location where it first appears.

As suggested by the reviewer, we moved the definition of primary vegetation (and cropland) in the last paragraph of the Introduction – lines 93-94.

-
- Lines 128-130: It would be important to have some examples of measures of abundance.

We thank the reviewer for pointing out this gap in our text. We have now added information of how abundance was measured and percentage of sites with abundance information for each type of measurement (lines 516-518).

-
- Line 270: The authors use “subsistence yields” as a proxy for “agricultural suitability.” The rationale seems to be that the former does not require “inputs, irrigation, and most human management.” If I understand correctly, this seems to be a rather narrow definition of “subsistence yields,” because even subsistence yields often require some external inputs or irrigation. I suggest the authors provide a more thorough explanation for using “subsistence yields” as a proxy for “agricultural suitability.”

We agree with the reviewer that in reality subsistence yields often involve some external inputs or irrigation. However, the term “subsistence yields” is used as a convenient shorthand in models of agricultural yields (the outputs of which we use here) for yield modelled under an assumption of no inputs, no irrigation and minimal human management. We are using this term with the same meaning. We added this information in the Methods section (lines 632-635).

The goal of using modelled subsistence yield in our models was to establish a baseline environmental suitability for a given crop. The need for such a baseline is explained on lines 625 - 632.

-
- Line 377 - Section “Statistical Analysis”: I would like to see the set of “equations” used. This would be a good way of summarizing the information that is, in fact, quite dense. These “equations” could simply be the instructions used in R to define the statistical models, or some schematic illustrations of how variables and parameters are related.

As suggested by the reviewer, we have added to the Statistical Analysis section of the Methods the R formulae of the initial models for the yield and conversion models (lines 741 – 760).

We have also replaced figure 1c with a new figure that summarises the structure of the statistical models.

The R equations of the selected models for each analysis are presented in the supplementary table S7.

The authors say that “to deal with overdispersion, we reran the species richness models substituting the Poisson distribution with a negative binomial distribution and dropping the random effect for site.” (my emphasis) Maybe this wasn’t necessary given that the model is a mixed-effects model (a consequence that is more clearly understood under the Bayesian approach). In these models there is hierarchical structure where the mean of the Poisson is being modelled with some distribution. For example, when the mean of a Poisson distribution is modelled by a gamma distribution, then the resulting distribution is a negative binomial. In other words, when using a Poisson with a mixed-model/hierarchical model, the overdispersion is automatically being taken care of (e.g., the Wikipedia: https://en.wikipedia.org/wiki/Negative_binomial_distribution) or McElreath, R.

(2020). Statistical rethinking: A Bayesian course with examples in R and Stan. Chapman and Hall/CRC). This may be somehow related to the authors' observation in lines 637-638.

We thank the reviewer for their point. Indeed, including in the model a random intercept representing each data point allows the model to work under the assumption that each data point is drawn from its own Poisson distribution with individual, unknown means. The means of these distributions can then be modelled separately. However, statisticians working within the frequentist framework highlight that this approach does not always perform best, especially at high levels of overdispersion, and that it is recommended to compare estimates of the observation-level random effect model with a model assuming a negative binomial distribution (without an observation-level random effect) ⁴. Therefore, we have decided to use both approaches to ensure that the models are robust to the choice of overdispersion treatment.

• Line 591 – Section “Biodiversity impacts and land conversion”: I suggest presenting the results of this section in a table.

We thank reviewer for this suggestion. We have now transferred the results from this text into an extended data table (Extended Data Table 1) and we have re-written this subsection of the Results section (L 229 - 257). Please, note that the conversion analysis has changed to account for biogeographical bias in the data point distribution, as raised by another reviewer.

• Line 692: Can you say that often “species with large global ranges” are generalists?

As suggested by the reviewer, we have added this detail on line 364.

• Line 711: I found the paragraph starting on line 711 difficult to follow. The authors start by discussing “wheat landscapes,” but then change to “maize and soybean landscapes” (line 719) and later to “rice landscapes” (line 725). However, the transition between these topics is not clear, and I initially assumed the authors were going to compare their impacts.

We thank the reviewer for pointing out a potential source of confusion here. This paragraph is discussing the impact of natural habitat on the relationship between yield and biodiversity, and compares the interaction between percentage of natural habitat and yield across the four crops. We have now made this clearer the first sentence of the paragraph, which now summarises the main point of the paragraph (L383- 385).

• Lines 726-728: I don't understand the conclusion: “This suggests that species richness and abundance increases with rice yield in landscapes with a high percentage of natural habitat is driven by narrow-ranged species.”

We have corrected and expanded the text describing the result that supports this conclusion. It is the RCAR responses to the interaction between rice yield and percentage of natural habitat that suggest that narrow-ranged species are benefitting in relative terms from yield increases in landscapes with high percentage of natural habitat (lines 396 - 402).

• Lines 741-742: The value 99.7% seems excessively high.

We thank the reviewer for pointing out this limitation of our models that we overlooked in the initial manuscript of our paper. Indeed, our models are predicting biodiversity levels when closing yield gaps based on novel combinations of variables while using log-links or log-transformed response variables. Therefore, it is likely that the extreme values are unrealistic⁵ We have now added this information to the discussion of limitations (lines 475 - 479) and have replaced the text indicated by the reviewer with a broader description of trends (lines 413-417).

• Line 785: There is a reference to straight lines (most likely correctly) and then a reference to Figures 4 and 5, but these do not depict straight lines. This is a little bit confusing because the relationship between the straight lines first mentioned and the results in the figures is not clear.

We apologise for the confusion. By straight lines, we meant the artificially straight border between the tropical and non-tropical areas. We re-phrased the text to avoid confusion (lines 467-468).

Figures:

• Figure 1a: Why do some “dots” have different color intensities (e.g., in South America)?

We thank the reviewer for pointing out this omission. The symbols in both 1a and 1b are transparent in order to make overlapping data points visible. We have now clarified this in the caption (lines 131-133) and added the transparency to the legend of the figure. Please note that the transparency of symbols is further reduced by overlap with sites that are in close proximity but not at the same exact location.

• Figure 1b: The relative size of the circles is too similar. Why are some points circumferences and others circles?

We have now increased the range of sizes of the circles. However, please note that we cannot increase the size of the circles beyond the current maximum because the individual sites would become difficult to distinguish in areas with many data points. We have also increased the minimum size of the circles so that they do not appear as points and create confusion. Information about transparency of symbols and its meaning was now added to the legend and caption of the figure (lines 131-133).

• Figure 1c: I don’t understand this plot. What is the meaning of the “axes”? What is the “grey square”?

We have decided to remove this figure due to the confusion it created. We have replaced it with a figure summarising data sources and variables included in our models (line 143).

• Figure 2: I see no need to show the green dots for “Primary vegetation in natural landscapes.” This is just a reference value that is implicit in the analyses. (The labels on the y-axis are not fully visible.)

We thank the reviewer for this point. We agree with the reviewer that “Primary vegetation in natural landscapes” is meant as a reference value in the analyses. However, we prefer to keep it in the figures to make the figures easier to interpret on their own. Moreover, the

absence of Primary vegetation in natural landscapes” might create confusion for the reader as the text refers to four categories of the land use – landscape variable.

As suggested, we have adjusted the labels on the y axes.

Supplementary Material: Supporting Figures:

- What is the meaning of the dashed red lines in figures S5-S12?

The dashed red lines in the figures S5-S12 (renumbered S5 – S14) represent the observed 0.5 quantiles in the y direction of the simulated residuals. The solid red line indicates the theoretical 0.5 quantile of the distribution of simulated residuals. We have now added this information to the captions of the supplementary figures S5 – S14.

-
- ATTENTION: Some of the figures have the same “number” (two figures S5, two figures S9, and two S14).

We thank the reviewer for pointing out this mistake. We have now corrected the numbering of the supplementary figures.

=====
Reviewer #2 (Remarks to the Author):

In their manuscript titled “Biome type and availability of natural vegetation dictate whether farmland intensification or expansion is worse for biodiversity”, the authors compare the impact of agricultural expansion and intensification scenarios on biodiversity, represented by metrics for species richness, total abundance and the relative community abundance-weighted average range size (RCAR).

In order to compare the scenarios, the authors assume a production increase target of +1% at each pixel that is obtained either by intensification or expansion.

The authors address an important and relevant research question at the global scale.

The paper is well written and basically uses appropriate data and methods, that could somehow be clearer presented.

We thank reviewer 2 for the positive comments made here and the constructive comments below.

General comments:

While the biodiversity analysis is well performed and explained, my main general concern is about the conceptual framework and underlying normative assumptions of the study.

For comparison of intensification and expansion, both scenarios aim at reaching the same production increase target of +1%. This seems arbitrary to me, and it is not explained why 1% is chosen.

We thank the reviewer for pointing out this gap in our text and we take this opportunity to explain this choice on lines 109-112 . Due the fact that as the expansion scenario involves land conversion (from primary vegetation to cropland, as justified on lines 114-119) which is proportional to the production increase and existing cropland in a given raster cell, the expansion scenario could lead to invalid proportion of land uses in raster cells with high percentage of cropland and low percentage of primary vegetation. For example, the proportion of primary vegetation could become negative or the proportion of cropland could

become higher than 1. In these cases, we decided that the reasonable approach would be to remove these cells from the analysis. We therefore chose a 1% production increase to minimise the number of raster cells that needed to be removed from the analysis. Importantly, while the assumed 1% production increase is an arbitrary choice, the direction of the results would not change depending on the assumed increase in production, although the magnitude of the differences between intensification and expansion would.

To illustrate these points, we created a supplementary figure with a production increase of 10% (Figure S32). The outcome in terms of whether expansion or intensification is the better option does not change, although the magnitude of differences increases strongly. However, this scenario leads to the removal of 5.1%, 9.1%, 4.5% and 1.7% of raster cells from the maize, soybean, wheat and rice analyses, respectively. We further tested the robustness of our conclusions to another production increase target and to different intensification patterns (L846-849), and we present the results in Figures S33-35.

Further, the assumptions underlying the two land use scenarios are very normative and I am afraid that the assumptions also largely determine the results. The authors write that their chosen scenarios are not realistic projections for the future, but rather capturing an extreme case of possible strategies to increase agricultural production. Nevertheless, there is no theory behind, what is driving agricultural expansion/intensification and where this is gonna happen.

We thank the reviewer for raising this point.

Here, we are not trying to understand the global impact of increasing agricultural production or predict realistic land-use change patterns. Instead, we are interested in comparing local biodiversity impacts of different strategies to increase production within every cultivated cell (L89-91 and L105-114). We now clarify in the text that this is justified by the high uncertainty of where land-use pressure will actually increase in the future⁶. Therefore, it remains relevant to explore the relative biodiversity impact of expansion or intensification within as many agricultural landscapes as possible (L 41-43). Specifying a certain percentage of production increase only serves as a way to keep scenarios comparable within a cell, in this following the principle of comparing alternatives that are matched in terms of total production^{7,8} (line 105-109).

As we explain above, the choice of 1% production increase is made to minimise the number of raster cells that are removed from analysis. More importantly, the choice of any particular production target does not modify the result of the comparison between expansion and yield increase, although it increases the difference between the two strategies. To make this point, we now perform robustness checks by calculating the biodiversity impacts of expansion and intensification at the level of every raster cell under different assumptions: a higher production increase target (Figure S32), a production increase target proportional to the local yield gap (Figure S35) and different intensification patterns within raster cells (increase yield by 2% on 50% of cropland – Figure S33, increase yield by 10% on 10% of cropland – Figure S34). All these analyses supported the key message of our analysis – that neither intensification nor expansion is better for biodiversity in every agricultural landscape.

Changing these assumptions could probably lead to very different results and conclusions, which affects the robustness of the approach.

As mentioned above, we have now performed checks of the robustness of our assumptions regarding production increase target and intensification patterns (Fig. S32-35). These checks showed that changing the assumptions does not change the main conclusion of our analysis: that neither intensification nor expansion are consistently better than the other in terms of biodiversity as an option for increasing agricultural production within a landscape.

As an example, expansion is only allowed into primary vegetation and takes place at each pixel up to the 1% production increase target in the comparison scenario. The 1% production increase target applies to every pixel. Why e.g. not implementing a global production increase target?

We thank reviewer 2 for this question, which gives us the opportunity to explain better our choice.

In terms of expansion options, cropland can indeed expand into pastureland or secondary vegetation instead of primary vegetation, resulting in lower biodiversity costs⁹. Given that the results of our analysis go against most academic literature in suggesting that expansion is not always more costly than intensification, we wanted to stress-test this conclusion against the highest biodiversity cost of expansion (L116 - 119). Moreover, most comparisons of intensification and expansion focus on conversion of primary or natural vegetation into cropland¹⁰⁻¹² (L114-115). Therefore, the structure of our models is aiming towards quantifying this particular type of agricultural expansion.

In terms of implementing a global production increase target, we did not do that because our aim was not to project global biodiversity change under a realistic scenario of production increase. Such a goal would be hampered by the high uncertainty of where land use pressure will increase in the future⁶. Instead, our goal was to explore the relative magnitude of the local biodiversity impacts of increasing production either through expansion or intensification within each raster cell. Considering these research aims, a hypothetical scenario with an arbitrary production increase fits our requirements of comparing alternatives matching in terms of total output. We edited the last paragraph of the introduction to better explain our goal for this analysis (L89-91 and L105-114)

Yields are increased uniformly on each pixel by 1%, unless a yield increase by 1% in regions with low yields (high yield gaps) increases absolute yields only very marginally. Therefore, closing yield gaps by a certain percentage value would have been more intuitive.

As we are interested to understand the balance of biodiversity costs of expansion and intensification within each raster cell, we chose to uniformly increase production by 1% because the direction of the difference between expansion and intensification at raster cell level would not change with a different production target (although the magnitude of the differences between intensification and expansion would change). To illustrate this point, we created a supplementary figure where the production increase is equal to 1% of the local yield gap multiplied by the area cultivated with the respective crop (Figure S35). As you can see, the broad patterns of whether expansion or intensification is the better option does not change, although the magnitude of the differences does change proportionally to the yield gap. Therefore, having different production increase targets for different cells would not modify the key message of our analysis.

It would also be more realistic to assume that yield increases are stronger in regions with high yield gap, due to the fact that yield gaps are already almost reached (and even above in some regions) in high productive regions.

We thank the reviewer for this comment. As we pointed out above, modifying the production increase target to be proportional to the local yield gap does not change the main outcome of our analysis (Figure S35), it would only lead to the removal of certain areas from the analysis. We wanted to avoid that because we wanted to calculate this balance of biodiversity impacts between expansion and intensification for as many raster cells as possible due to high uncertainty in land-use change projections⁶ (lines 41-43). Even in areas where yields exceeded production limits in the EarthStat data for the year 2000, yields might still increase in the future due to technological and management advances.

Finally, you conclude that neither intensification nor expansion is consistently better for biodiversity in context of the land sparing land sharing debate. However, the intensification approach does not consider that less land is used for agriculture and could therefore be given back to natural ecosystems, because in your approach, you uniformly intensify all cropland by 1% within a pixel (at landscape scale), but also regionally or globally, which to my understanding is not land sparing, which is a compaction of used land that goes along with a renaturation of marginal lands, not only an intensification (compare e.g. Schneider et al. 2022, <https://doi.org/10.1371/journal.pone.0263063>).

We thank reviewer 2 for raising this point. When designing our intensification and expansion scenarios, we relied on Balmford (2021, p.80)⁷, which notes “The sharing/sparing framework also allows the exploration of a continuum of solutions involving yields and areas under nature which are intermediate between extreme sharing and extreme sparing [...]”. In this definition, expansion and intensification in the context of land-sharing/sparing does not necessarily imply a reduction in cropland. Instead, we chose to model intermediate expansion and intensification scenarios because modelling scenarios where land is given back to natural ecosystems would require additional assumptions and models about the type of restoration taking place, and the speed and level of biodiversity recovery on the land taken out of production, which are highly uncertain and debated^{13–15}. We add this clarification to the manuscript on lines 439-442.

In terms of spatial scale, comparing intensification and expansion at the raster cell level across all global agricultural landscapes is justified by the high uncertainty of where land-use pressure will increase in the future⁶ and it is in accordance with the suggested sparing/sharing scale (patch size of at least 1-10 km², larger in homogeneous landscapes but not larger than 10⁴–10⁵ km²) suggested by Balmford (2021, p. 81)⁷.

In case of expansion, more land is used and no land is taken out of production and restored to natural ecosystems, which is a major reason why land use expansion in average is worse for biodiversity, at least if you look at this at the landscape scale. Accordingly, linking your approach to the land sparing/land sharing debate is difficult, and therefore, in my opinion, your conclusion is biased in that regard.

As we have indicated above, we relied on Balmford (2021, p.80)⁷, who defined land sharing/sparing as a continuum. We now explain in the manuscript that we did not consider all possible alternatives within this continuum (L439-442 and L820-823). Instead, we chose to model intermediate expansion and intensification scenarios because modelling scenarios

where land is given back to natural ecosystems would require additional assumptions and models regarding biodiversity restoration¹³⁻¹⁵ (L435-438).

We have also now added several supplementary figures showing that our conclusions are robust to the choice of production increase targets and intensification patterns (Fig. S32-35).

What you would have to do is to intensify in each pixel e.g. by 10%, then reducing the used land to reach the 1% production increase target and restore natural ecosystems on the freed up land. Would that be possible?

We thank reviewer 2 for this suggestion. We assume here that the reviewer suggests a scenario that assumes short-term and complete restoration of biodiversity on the freed land, an approach that we consider problematic due to the uncertainties of restoration¹³⁻¹⁵. Otherwise, this approach would involve biodiversity recovery models that are outside the scope of this study.

We did run the analysis as suggested by the reviewer under an assumption of complete restoration on the freed land. We include below the resulting figure. For this comparison, we kept the expansion scenario as in the initial draft i.e., increase production by increasing cropland area by 1%; for the intensification scenario, we increased yield by 10% and calculated that 91.8% of the cropland area will be sufficient to achieve a 1% overall increase in production. We then “restored” 8.2% of the cropland to primary vegetation and its associated biodiversity. As can be noticed from the figure below, the patterns of whether intensification or expansion are better for biodiversity does change in some cases (RCAR in wheat landscapes) but the central conclusion, that neither intensification nor expansion is consistently better across crops and biodiversity metrics, still stands. Considering the problems this approach raises in terms of unrealistic biodiversity gains from restoration and the fact that it would not change our main conclusion, we prefer not to include this in our study but we add this figure below.

The difference in biodiversity metrics when comparing the land expansion and intensification scenarios. The colour range symbolises areas where increasing total production by 1% through cropland expansion is better for biodiversity (blue hues) and areas where increasing total production by 1% through intensification, cropland contraction and restoration on 8.2% of cropland is better for biodiversity (red hues). RCAR (i, j, k, l) increases and decreases were considered to be the negative and positive outcomes for biodiversity, respectively. We removed 0.9%, 2.3%, 0.5% and 0.2% of raster cells from the maize, soybean, wheat and rice analyses, respectively, due to invalid land use coverages (negative or over 100% coverage for a single land use type) resulting from the 1% expansion scenario. Please note that restored cropland has been converted to primary vegetation in this exercise, despite it not fitting the definition from the main text, in order for our models to be able to project these results. Our models only contain 2 land-uses: cropland and primary vegetation.

Data and Methods: It is not easy to follow the methodology at first reading due to the many different data sets. I had to read several sections twice. An overview is missing, which would help a lot to quickly understanding the used data and methods. Therefore, I suggest showing a schematic figure for both, the yield-biodiversity models and the land-conversion models, giving an overview on the applied data and how they were used.

We thank the reviewer for this comment. As suggested, we have now added a summarising figure for the yield and land conversion data and models (Figure 1c).

The time period the authors target their scenario is approximately the year 2000 and not a

future projection, considering climate change, etc., which would make the study more interesting.

We agree with reviewer that considering climate change would indeed make the study more interesting and it might be something that we could pursue in the future. However, that is outside the scope of this paper and space limitations do not allow us to pursue this additional line of research.

Instead of using potential yield estimates by Mueller, that are based on observed yields within AEZs, you could use potential yields, simulated by crop models, which take spatial heterogeneity and future climate change impacts into account. They tend to be higher than observation based approaches, especially in developing countries.

We have indirectly used the potential yield estimates by Mueller (by using the estimated yield gaps from the respective publication) because they are consistent with the EarthStat estimates. Moreover, potential yields datasets would come with modelling assumptions that would add a new layer of complexity in the interpretation of our results.

We are also focused in this study on understanding the biodiversity effect of yield as driven by agricultural management. Including climate change in the yield gap calculation would, again, complicate the interpretation of our results.

In the discussion, I would additionally expect one sentence saying that also other ecosystem services rather than biodiversity must be considered in an integrative approach, such as carbon storage and water infiltration, etc.

We thank the reviewer for this suggestion and we agree that this was an important omission in the discussion. To address this point, we have now added a sentence on lines 489-490.

I am wondering if you consider any kind of maximum expansion potential, which limits agricultural expansion mainly in bread basket regions, because available areas for expansion are very limited here.

We agree with the reviewer that this would be an interesting constraint to be added to scenarios. However, this condition would address a narrower research question (what are the biodiversity impacts of agricultural management taking into account land availability constraints) from the one that we asked in this study (how do intensification and expansion compare within agricultural landscapes globally). Unfortunately, due to space constraints, we cannot consider this aspect in our analysis.

While the study gives interesting new surprising findings and a new view on the effects of land intensification on agriculture, I am still not fully convinced that these findings are robust and not a result of the assumptions and the conceptual framework of the approach. This needs to be clarified. Therefore, I suggest major revisions of the manuscript.

We thank reviewer 2 for their thoughts and suggestions. We hope we have managed to justify and support our assumptions and conceptual framework.

Minor comments:

Title:

I suggest to use the term “cropland” instead of “farmland” in the title, because farmland could include pasture areas or rangeland which is not considered in your approach.

We thank reviewer 2 for this suggestion, with which we agreed. We replaced “farmland” with “cropland” in the title of the article.

Abstract:

While you quantify the effect of intensification on biodiversity in the abstract by a median species loss of 11%, you just say that land conversion is associated with significant biodiversity loss”. I suggest to give quantitative results for both aspects for comparison, which would improve the significance and informative value of the abstract.

We thank the reviewer for pointing out this inconsistency. We have now reformulated the abstract to include quantitative results for the conversion analysis as well (lines 21-24).

Figures:

Figure titles are used instead of figure subtitles, which confused me.

We are unsure what the reviewer refers to here. In editing the captions of our figures, we followed the formatting instructions of Nature Ecology and Evolution, which say: “Include a brief title for each figure with a short description of each panel cited in sequence.” Our captions include therefore a title (lines 121-122, 146-147, 161, 201, 214-215), in bold letters, followed by the description of each panel (lines 122-140, 147-152, 161-164, 201-204, 215-217).

Maps are not well illustrated. I’d suggest to use a global frame and cut at -60° South (showing all global maps without Antarctica).

We thank the reviewer for this suggestion and we have now changed the maps to leave out Antarctica.

Specific comments:

Ln 25: “yield increases also lead to” instead of “yield also lead to ...”

We thank reviewer 2. We made the correction as suggested (L25).

Ln 42: Which has already been done by some other studies, e.g. Zabel et al. (2019). <https://doi.org/10.1038/s41467-019-10775-z>

We thank reviewer 2 for highlighting this publication. We are citing it now on line 45.

Ln 51: , whereas

We thank reviewer 2. We made the correction as suggested (L52).

Ln 66-70: This is an important aspect, referring to the discussion on sustainable intensification, which could also be mentioned here as a buzzword. Nevertheless, sustainable intensification is not considered in your approach, which I assume, would impact

on the comparison between expansion and intensification quite drastically, also to the authors' conclusions. That should also be added to the discussion. Also, I would be interested in an explanation, what kind of intensification this data driven statistical approach implies in different regions and how this could impact on biodiversity?

We thank reviewer 2 for the suggestion of using “sustainable intensification” here. Considering the existence of several terms, such as sustainable intensification and ecological intensification, and their evolving definitions, we wanted to be more specific here, hence the reference to relying on ecosystem services and benign technologies.

As suggested by the reviewer, we now acknowledge in the Discussion section that sustainable intensification and, in general, changes in management practices (lines 453-455), might change the biodiversity trends presented in our paper. We specify that the kind of intensification captured by our analysis is most likely to be the conventional type, which was the most widespread type in 2000, and probably currently, and characterised by increased inputs (lines 449-452).

Ln 211-213: At least for rice, I assume that rainfed systems are also not used by subsistence farmers, but honestly I don't know.

We use the term “subsistence yield” as defined and used in the yield modelling literature, which considers agriculture modelled without irrigation and inputs as subsistence agriculture (now explained on L632-635). Therefore, our goal was not to represent through these data actual subsistence yields but was to estimate yield at minimum level of human management.

Ln 364-367: This is an important limitation of this study, because usually cash crops might be used at much higher intensity level than other crops. Also, I guess that there is a huge difference between crops. I was wondering if ‘cropping system’ is the right term here? Maybe, ‘across different crops’ might be more accurate?

We agree with reviewer and we added this point in the discussion of the limitations of our analysis (L 463-465).

We also replaced ‘cropping system’ with ‘across different crops’, as suggested by the reviewer (L713).

Ln 452: blank before full stop.

We thank reviewer 2. We deleted the space before full stop.

Ln 476-478: This is an important assumption for understanding the whole study and I have been missing this information while reading the chapters before. I was always wondering, where expansion and intensification takes place and by how much you close yield gaps, etc. Now this is explained here. You expand and close yield gaps to reach a uniform production increase by 1% for each pixel and each crop. Maybe mention that already earlier, maybe end of introduction where you describe the general study framework.

We have now added this information in the introduction and we have clarified that closing yield gaps and the 1% increase in production are two different analyses (lines 87-109).

Line 484: If you increase the area by 1%, you assume homogeneous yields within a pixel, which may not be correct. I guess this overestimates yields, because usually, the best land is already in use and cropland is expanded into less productive land, at least in human modified landscapes. Another issue might be that expanded land often is used at lower intensity, while you assume the same yield gap than for the rest of the area within the pixel, which may not be the case. These assumptions should be addressed and/or discussed.

We agree with reviewer's point that expansion might take place in areas that are less suitable for the respective crop than existing farmland within the pixel, which might change the management interventions required to obtain the average yield of the pixel. We added this point on lines 817 - 820.

Regarding the intensity level of newly converted land, our expansion and intensification scenario are not intended to be realistic scenarios but illustrative scenarios of different yield increasing paths and their biodiversity impacts. While in reality newly converted farmland might be managed at a different intensity, we were interested in the hypothetical of what would be the biodiversity impact if farmland were expanded and managed at the average intensity of the respective pixel. We now explain in more detail our reasoning on lines 105-114.

Ln 488: In the previous part of the article, including the abstract, you mention closing the yield gap. Also Figure 4 is showing the effects of closing yield gaps (not saying by how much, I assume fully closing). However, increasing yields uniformly by 1% is s.th. else than closing yield gaps by 1%. You do not explicitly say that you close yield gaps by 1%, but at least reading the previous parts, it gives the impression and therefore, the text is somehow confusing in this context.

We thank the reviewer for pointing out this lack of clarity. We have now modified the last paragraph of the introduction to make it clearer that we are talking about 2 different analyses: fully closing the yield gaps on existing cropland (Figure 4) and a pair of scenarios on increasing overall production by 1% through either expansion or intensification (Figure 5) (lines 87-114).

Another question relating to these very small intensification levels in regions with large yield gaps is the question if the statistical approach is sensitive enough to capture the impacts on biodiversity of such small absolute yield changes? Especially for subsistence farmers, inter-annual yield variabilities are much higher than 1% meaning that a yield increase of 1% cannot significantly be related to changes in management.

We agree that interannual variabilities are likely to be much higher than 1% and the yield in any particular year can be quite different from other years. This is why, for example, EarthStat data are averaged across 5 years. By 1% in yield, we are referring to average yields. If we imagine possible yields at a given management level, in any location, across time to be represented by a normal distribution, what we use in our analysis is the average of that distribution of all possible yields. What gets changed in the intensification scenario is the average of that distribution of possible scenarios, which would change with a change in

management (and ignoring changes driven by climate change for our purposes, as explained above).

We now explain on lines 109-112 why we choose 1% as opposed to a higher increase in production. Choosing a higher increase in production would also lead to the same key conclusion, that neither expansion nor intensification are always the better solution. To demonstrate our point, we are also doing an analysis for 10% increase in yield (Figure S32). As can be noticed, the patterns of whether expansion or intensification are stronger, are identical with Figure 4 in the main text but the magnitude of the differences is higher.

Ln 592-602: Not clear at first reading: Does this refer to a land conversion of all primary vegetation or to the +1% production increase scenario?

These are the results of the land conversion models, not of the scenarios (question 1 on lines 85-86). To avoid confusion, we changed the title of this section to "Biodiversity impacts of local and landscape habitat conversion".

Appendix Figure S23. Source missing.

We thank the reviewer for pointing out this omission. We have now added the source of the data.

=====
Reviewer #3 (Remarks to the Author):

Opinion on the paper by Silvia Ceaușu, David Leclère, Tim Newbold: Biome type and availability of natural vegetation dictate whether farmland intensification or expansion is worse for biodiversity

Description and overall assessment

This global study examines the biodiversity effects of land conversion, and of two major strategies, namely increasing yield on the same land area, or expanding farmed area without intensification. These questions have been explored before. For instance, in a recent study by Jaureguiberry et al (2022, Science Adv), they found that land/sea conversion is the primary cause of biodiversity loss. The two major strategies, known as the "land sharing / land sparing dilemma," have also been the subject of numerous articles, many of which have concluded that there is no easy answer because it depends on the context. Therefore, I have some doubts regarding the novelty of these results. Nevertheless, it utilized quantitative ecology data from a diverse array of taxa, indicators and locations, thus, being probably the most comprehensive study, utilizing several global database/estimates.

We thank the reviewer for the suggested literature. As the reviewer pointed out, our goal was to provide a thorough analysis of intensification and expansion relying on the best available data. We also believe that our results are novel because some studies, such as Jaureguiberry et al 2022¹⁶, group intensification and expansion together when assessing the biodiversity impacts of land-use change, and therefore do not address the comparison between these two options for raising agricultural yields. Other studies of the biodiversity impacts of intensification have concluded that expansion/land conversion is more costly for

biodiversity than intensification^{7,12}. Our results show that this depends on context, which has been shown before at smaller scales but not globally.

Major comments

This study is extensive, with Figure 3 alone providing a broad variety of results to analyse, like why are there many relationships (like for soy and RCAR), while none in others (wheat and RCAR). The article analysed biome, crop, biodiversity indicators, landscape context, land use, and their interactions. However, only a limited number of results were emphasized and described in the paper.

We thank the reviewer for this comment and for highlighting the potential for confusion here.

Figure 3 summarises the relationship between biodiversity and yield as resulting from the model selection process described in the Methods section (lines 774-776). For some models, the interactions between yield and each of the variables biome type, percentage of natural vegetation and land use proved to be significant, and were, therefore, kept in the final models (e.g., RCAR model maize yields). In the case of other models, none of the yield interactions or yield on its own was selected as significant in the final model (e.g., RCAR model for wheat yield). We have now modified the caption of the figure by moving some of the key information earlier in the text (lines 164 - 172) and rephrasing the description of the significance of the different colours and line types.

Regarding the difference between the number of variables included in the analysis and those discussed in the text, this is due to the fact that the relationships that are the focus of the study could be impacted by many other aspects of the environment. All variables included in the models were included because they were considered relevant to understanding the relationship between biodiversity and yield (as explained, for example, on lines 607-609, 620-623, 625-631, 660), which is the focus of the study and, to a certain extent the relationship between biodiversity and natural habitat, which is also important to the conclusions drawn here. However, the text of the Results and Discussion sections focuses on the relationships that are answering the research questions on lines 85-91.

I find it difficult to fully endorse the prioritization of yield over factors such as fertiliser consumption, particularly when the focus is solely on biodiversity benefits and socio-economic aspects are not taken into account in the study. Productive land can achieve high yields without relying heavily on agrochemicals, but low quality areas may experience low yields despite using large amounts of fertilizers. Consequently, in the second scenario, the yield is low, but the negative impact on biodiversity is greater.

We thank the reviewer for this comment. Indeed, focusing on yield will obscure specific aspects of land management, a limitation that we now include more explicitly in the paragraph on limitations (L455-457). Capturing agricultural intensity globally is a difficult challenge and all approaches are likely to have advantages and limitations¹⁷, a fact we now acknowledge on lines 59-60. For the purpose of our study and our research questions, we considered that the advantages of focusing on yield, as described on L63-70, outweigh the limitations, which we describe on lines 447-460. For example, this approach allows us to model expansion and intensification for the same level of output, which is an important principle in comparing alternatives^{7,8}.

Biotic homogenisation is a measure of community composition, restricted to spatial context. Note, however, that this not characterise the species in the community, as range sizes are influenced by human activities, e.g. via extirpation of suitable habitats.

We thank reviewer 3 for this comment. We agree that current range sizes are already influenced by humans. Indeed, we are not using biotic homogenisation to characterise species in the community, we are using it as a measure of community composition, as specified by the reviewer. As a metric of biotic homogenisation, we use relative abundance-weighted community-average range size (RCAR), which measures how widely or narrowly distributed are the species in a community, on average, as captured by the data sources¹⁸. The RCAR metrics that we use is based on occurrence data from the GBIF database (GBIF.org, 2015), which likely captures a least some of the range changes due to human activities.

Why 1% increase in total production was targeted? 5% or 10% increase would be also interesting. As we now that c 800 million people are starving and we are 8 billions, a 10% would be realistic.

We thank the reviewer for this point. The goal of our scenarios was not to be realistic, a point made clearer now on lines 109-114, but rather to compare projected biodiversity impacts of expansion and intensification within agricultural landscapes. Moreover, any increase in production that we would choose would lead to wider differences in biodiversity impacts between expansion and intensification but the same key conclusion: that neither expansion nor intensification is always better for biodiversity.

To show this is the case, we have now added a supplementary figure (Figure S32) for which the increase in production is 10% of the initial production level. The areas in which expansion or intensification are the better option are the same as in figure 5 in the main text although the differences are more pronounced. As we now justify on lines 109-112, we chose 1% to minimise the number of raster cells eliminated from the analysis due to invalid land use proportions (e.g., cropland occupying over 100% of a raster cell or primary vegetation coverage below 0%) when expanding cropland in the expansion scenario.

On the other hand, however, if food system would be effective (no waste, no overconsumption /c. 900 million live with obesity/, etc.), what biodiversity gain would be if 1%, 5% or 10% decrease in yield will be possible?

We agree with the reviewer that there are important ways in which our food systems could become more effective and our diets healthier. We acknowledge this fact on lines 493-494. However, this particular research question and scenario are outside the scope of this paper and would involve additional analysis (e.g., assessing biodiversity impacts of different forms of restoration in the case of farmland contraction).

I enjoyed the presentation of study limitations (from line 773), as it listed several of my concerns I noted while reading the manuscript, like missing of local scale, data/database quality, straight line across the maps. So this is a valuable part of the paper, clarifying its

uncertainties. But these exist. In addition, the models not fully fit assumptions. Thus, results need some caution when interpreting.

We thank the reviewer for this comment. We agree that the paragraph on limitations in the Discussion section is crucial to interpreting our results. We have updated it now with a few more points raised by reviewers (lines 447 – 479).

Further comments:

What is closing the yield gap? Define.

We have now defined yield gaps in the last paragraph of the Introduction on lines 87-88.

Line 42: From this point, the introduction is on land sharing/sparing. However, these terms are ignored from the abstract, making a little bit of mismatch between these sections.

It was not our intention to focus the Introduction on land sharing/sparing and we would like to respectfully draw the attention of the reviewer to the ways in which the text addresses several other issues. For example, the paragraph that the reviewer refers to ends with a discussion of studies of local community metrics in relation to yield, which are not land sharing/sparing studies because these usually use metrics of population density⁷ (L53-58). We would also like to point to the next two paragraphs after the one the reviewer highlights, the first of which focuses on other studies of land-use intensity that use very different approaches from land sharing/sparing studies (L59-70) whereas the second paragraph discusses different biodiversity metrics used in studies of biodiversity impacts of land use (L71-81). Many of the studies cited in these two paragraphs (for example: Leclère et al.¹⁹, Semenchuk et al.²⁰, Attwood et al.²¹, Beckmann et al.²², Newbold et al.⁹) are not about land sharing/sparing.

Although the land sharing/sparing debate is relevant to our research questions and analysis, and is addressed in Discussion and Introduction, our study fits within a wider discussion on agricultural intensification and expansion^{9,22,23}. Therefore, we do not think that it is necessary to specifically refer directly to land sharing/sparing in the abstract, especially taking into account space limitation and the need to define these terms. We do however mention land sharing/sparing indirectly by referring to studies measuring species persistence (L14).

Line 56-57. „Neither species persistence or community metrics response to yield increases were assessed globally” it is true but, Martin et al (2019, Ecol Letters) published a large study on yield and landscape at the European scale.

We thank the reviewer for directing us towards this paper. The particular lines referenced by the reviewer refer to biodiversity responses to yield, which Martin et al (2019) does not address directly. However, the paper is highly relevant and we cite it on line 53 in the Introduction and on line 357 in the discussion.

Line 97-98: why 30 and 70% are the shares?

We thank the reviewer 3 for the question. We rely here on Andren(1994)²⁴ who shows that ~30% and ~70% of suitable habitat in a landscape represent important thresholds in how habitat loss and configuration of habitat patches interact to lead to biodiversity loss. We now

moved and expanded the explanation for the choice of percentages from the Methods to the Introduction on lines 99-101.

Line 123: where are Table S1-2-3? Not elegant to start with Table S4.

We thank the reviewer for pointing out this mistake. The supplementary tables which we should have referenced here were S1-S5.

Figure 1. Looking at the map, I have the impression that the distribution of sites is aggregated, that is similar sites are clustered spatially. E.g. natural localities in natural landscapes are in SE Asia and Oceania, while Europe is the opposite. Thus, a conclusion that biodiversity decline is due to human ecosystem modifications can also be a spatial effect, that is the "treatments" are in different biogeographic regions. How to separate geography and human influence variables?

We thank the reviewer for making a great point here that we have not addressed sufficiently in the initial draft of the manuscript. In our yield models, we did include an interaction between yield and biome to account for the potential different impact of yield within different biogeographic regions. We now include also an interaction between the combined land use-landscape variable and biome type in the conversion models.

In addition, the clustering of points could also be driven by the different land use histories. To also control for this potential source of bias, we also added duration of significant human landscape modification and its interaction with the combined land use-landscape variable to the initial set of variables of the conversion models.

Based on the model selection process, the interaction between the combined land use-landscape variable and geographic region is significant for abundance and RCAR. The interaction between the combined land use-landscape variable and duration of significant human landscape modification is significant for species richness and RCAR. The new results are now presented (Figure 2, Extended Data Figure 1 and 2, lines 230 - 257) and discussed (lines 340 - 357) in the new draft.

There are also pale blue circles (in Chile and Argentina, Japan, NE America). Is this a typo, or has some meaning?

We have omitted to mention that the circles in Figure 1A and 1B are transparent and their opacity is indicating the presence of different number of sites at the same location. We modified the caption (lines 131 – 133) and the legend of the figure to include this information.

1/c figure. I cannot understand it...

We have now removed this figure and replaced it with a figure that summarises the data use across the statistical analyses.

Fig.2. Increase the distance between a) and b), otherwise there is overlap in Y axis legend.

We thank the reviewer for pointing out this error. We have now increased the distance between the panels of the figure so the titles of the y-axes do not overlap anymore in Figure 2.

Fig.2. On a figure with values and error bars, the reader will look for significances, which are missing from this figure, thus, it is not easy to understand where are the differences or similarities?

The significance of biodiversity change relative to primary vegetation in natural landscapes can be assessed by the overlap between the bars representing the 2.5th and 97.5th percentiles (95% confidence interval) with the line marking 0% change. We now highlight this information more clearly in the caption of the figure (lines 154-158).

Differences and similarities between the different categories other than primary vegetation in natural landscapes can be assessed based on the degree of overlap between the respective confidence intervals as our models do not provide information on the significance of the comparisons between the rest of the land use-landscape categories.

Fig.3. It has an extreme amount of information, really need time to digest it. Interesting to see that two graphs are empty, while others have a lot relationships plotted.

The number of relationships plotted depends on whether yield was selected in the final model and whether its interactions with land use, biome type or percentage of natural vegetation were significant. If all these three interactions were significant, we plotted a total of 8 different trends. We have also edited the figure caption to make it clearer by moving some of the important information closer to the beginning (lines 164 – 172).

Also interesting to see that abundance and RCAR show similar trends in maize and soy, indicating more individuals in more homogenised communities  a potential threat of invasions parallel with yield increase?

We thank the reviewer for this great point. We now mention the potential risk of invasions on lines 360-363.

Fig.3. line 538. This was 70% versus 30% above (line 97-98). Why different numbers?

The division between landscapes with 70% versus 30% natural vegetation is the classification for defining the combined land use-landscape variable that is used in the conversion models (L 99-101). The 85% and 15% natural vegetation are just used for illustrative purposes to show effects at high and low percentage of natural habitat. We have now made that clearer on lines 168-170.

In figure 3 several relationships show only one curve, that with the most data points. It ease illustration and understanding on a complex figure, but are the other relationships presented

somewhere? Did not the reader miss some important results?

We do not believe that the reader missed important results because the unrepresented trends are always identical as those presented in the figures, with the difference being only in terms of intercept (because relevant interaction terms were not selected in the final models).

To support this point, we have now included in the supplementary material Figure S26 that includes all curves for the models in which yield was significant and in which any of the variables land use, biome, and percentage of natural habitat were significant, including on their own.

Line 604: I would not say most often, as from the four crops it is true for two (missing in soy, and contrasting in rice).

We removed "most often" from this sentence.

Line 606: typo: space is missing.

We thank the reviewer for pointing out the mistake. We added the space where it was missing.

Line 645: Yes, but the blue colour (increase of richness) is rather clearly dominates tropical areas, like S Amazonia, and SE Asia. So the truth that increase in yield in the tropical areas has both benefits and costs on richness, while mostly costs in the temperate zone.

We thank the reviewer for pointing this out and we modified the text to highlight this point. This sentence has been edited to "The decrease in species richness is spread across both tropical and non-tropical areas while the increase in species richness is concentrated mainly in tropical areas (Figure 4 a), likely due to positive rice yield effects in landscapes with high percentage of natural habitat (Figure S29)." (L 308-311)

L 679-680: The role of landscape context was multiple times published see e.g. Batáry et al. (2011, Proc R Soc).

We thank the reviewer for pointing this out. We now acknowledge this fact and cite Batáry et al. (2011) on line 347.

L763: Others concluded that multifunctional landscapes are needed, a combination of sharing and sparing lands (Grass et al 2021, Adv Ecol Res).

We thank the reviewer for suggesting this reference. We have now added this reference on line 434 in the same paragraph.

Supplementary tables: there was probably some converting problem, a few letters were misplaced by „?”.

We thank the reviewer for drawing our attention to this converting problem. We will verify the documents at submission and contact the editor if we notice any issues.

Explain what is SR as this acronym was not used before.

We have now added the explanation for SR (species richness) in all the relevant supplementary tables captions. We have also added an explanation for RCAR - relative abundance-weighted community-average range size.

RCAR 55km is also new name (it was RCAR before).

We have now replaced RCAR 55km with RCAR.

Table S7, first column, correct column width to have „abundance” in one line.

We adjusted the column in table S7.

Supplementary figs: Fig S23: what is the dimension of the scale? %?

We have now added in the caption the unit of the scale, which is tonne/ha.

References

1. Hoskins, A. J. *et al.* Downscaling land-use data to provide global 30 "estimates of five land-use classes. *Ecology and Evolution* **6**, 3040–3055 (2016).
2. Williams, J. J. & Newbold, T. Vertebrate responses to human land use are influenced by their proximity to climatic tolerance limits. *Diversity and Distributions* **27**, 1308–1323 (2021).
3. Fleishman, E., Nally, R. M. & Fay, J. P. Validation Tests of Predictive Models of Butterfly Occurrence Based on Environmental Variables. *Conservation Biology* **17**, 806–817 (2003).
4. Harrison, X. A. Using observation-level random effects to model overdispersion in count data in ecology and evolution. *PeerJ* **2**, e616 (2014).
5. Altman, D. G. & Bland, J. M. Generalisation and extrapolation. *BMJ* **317**, 409–410 (1998).
6. Prestele, R. *et al.* Hotspots of uncertainty in land-use and land-cover change projections: a global-scale model comparison. *Global Change Biology* **22**, 3967–3983 (2016).
7. Balmford, A. Concentrating vs. spreading our footprint: how to meet humanity's needs at least cost to nature. *Journal of Zoology* **315**, 79–109 (2021).
8. MacKay, D. J. *Sustainable Energy-without the Hot Air*. (Bloomsbury Publishing, 2016).
9. Newbold, T. *et al.* Global effects of land use on local terrestrial biodiversity. *Nature* **520**, 45–50 (2015).
10. Phalan, B., Onial, M., Balmford, A. & Green, R. E. Reconciling food production and biodiversity conservation: land sharing and land sparing compared. *science* **333**, 1289–1291 (2011).
11. Macchi, L. *et al.* Trade-offs between biodiversity and agriculture are moving targets in dynamic landscapes. *Journal of Applied Ecology* **57**, 2054–2063 (2020).
12. Kehoe, L. *et al.* Biodiversity at risk under future cropland expansion and intensification. *Nat Ecol Evol* **1**, 1129–1135 (2017).
13. Atkinson, J. *et al.* Terrestrial ecosystem restoration increases biodiversity and reduces its variability, but not to reference levels: A global meta-analysis. *Ecology Letters* **25**, 1725–1737 (2022).
14. Crouzeilles, R. *et al.* A global meta-analysis on the ecological drivers of forest restoration success. *Nat Commun* **7**, (2016).
15. Benayas, J. M. R., Newton, A. C., Diaz, A. & Bullock, J. M. Enhancement of biodiversity and ecosystem services by ecological restoration: a meta-analysis. *science* **325**, 1121–1124 (2009).
16. Jaureguiberry, P. *et al.* The direct drivers of recent global anthropogenic biodiversity loss. *Science Advances* **8**, eabm9982 (2022).
17. Kuemmerle, T. *et al.* Challenges and opportunities in mapping land use intensity globally. *Current Opinion in Environmental Sustainability* **5**, 484–493 (2013).

18. Newbold, T. *et al.* Widespread winners and narrow-ranged losers: Land use homogenizes biodiversity in local assemblages worldwide. *PLoS biology* **16**, e2006841 (2018).
19. Leclère, D. *et al.* Bending the curve of terrestrial biodiversity needs an integrated strategy. *Nature* **585**, 551–556 (2020).
20. Semenchuk, P. *et al.* Relative effects of land conversion and land-use intensity on terrestrial vertebrate diversity. *Nat Commun* **13**, 615 (2022).
21. Attwood, S. J., Maron, M., House, A. P. N. & Zammit, C. Do arthropod assemblages display globally consistent responses to intensified agricultural land use and management? *Global Ecology and Biogeography* **17**, 585–599 (2008).
22. Beckmann, M. *et al.* Effects of conventional land-use intensification on species richness and production: A global meta-analysis. *Global Change Biology* **25**, 1941–1956 (2019).
23. Egli, L., Meyer, C., Scherber, C., Kreft, H. & Tschardt, T. Winners and losers of national and global efforts to reconcile agricultural intensification and biodiversity conservation. *Global Change Biology* **24**, 2212–2228 (2018).
24. Andren, H. Effects of habitat fragmentation on birds and mammals in landscapes with different proportions of suitable habitat: a review. *Oikos* 355–366 (1994).

Reply to Reviewers' Comments

Geography and availability of natural habitat determine whether cropland intensification or expansion is more detrimental to biodiversity

Reviewer #1:

Remarks to the Author:

I congratulate the authors by the changes.

Thank you!

Two small details:

First, I think the legend of Figure 1 is missing as well as figure 2.

The legend for Figure 1a and 1b is included at the bottom of the two panels (a and b) and above figure 1c.

The legend of figure 2 is in the upper right corner of the figure.

Line 503, I think there is a comma missing between "cropland" and "pasture" (otherwise it seems that "cropland pasture" is a class).

We have now added the comma as suggested by the reviewer – line 395.

Reviewer #2:

Remarks to the Author:

The authors provided a detailed reply and extensive revisions on all points. Most importantly they have addressed my scepticism regarding the conceptual framework of the approach and that normative assumptions might influence the results. Therefore, additional analyses were carried out to demonstrate the robustness of the results. Other aspects, that haven't been addressed before, are now added to the discussion. I am a little bit surprised that some of the explanations given to my comments and figures are not included to the discussion, or added to the supplementary, since I think they would have been interesting for other readers as well. The paper structure improved a lot, especially the revisions at the end of the introduction help a lot understanding the study.

We thank the reviewer for the positive comments. As far as we can tell, the only element from the previous round of reviews that was not included in the main text or supplementary information was the figure showing the results of an additional intensification scenario suggested by the reviewer that involved a 10% yield increase on 91.8% of the cropland area and "restoration" to primary vegetation of the remaining 8.2% of cropland area. We have now included this figure as supplementary figure S36. We refer to this additional intensification scenario on line 227 (in the Results section) and 653-654 (in the Methods section).

In Line 344, the authors write that the results of their study are consistent with previous studies on local land-use change and landscape composition effects. However, there is also a bunch of

studies that come to different conclusions, maybe at different scale and also by using different approaches. Nevertheless, I think it is important and also interesting to place the results of this study in the context of these studies (see below) and discuss differences and implications. You could refer to these studies:

Folberth et al. (2020) (<https://doi.org/10.1038/s41893-020-0505-x>), Schneider et al. (2022) (<https://doi.org/10.1371/journal.pone.0263063>), and Beyer et al. (2022) (<https://doi.org/10.1038/s43247-022-00360-6>) e.g. found that global cropland could almost be halved within planetary boundaries by sustainable intensification. On the other side, Schneider et al. (2024) (<https://doi.org/10.1038/s41893-024-01410-x>) e.g. found that cropland expansion resulted in very high trade-offs regarding biodiversity and GHG emissions however using a very different approach. Foley et al. (<https://doi.org/10.1038/nature10452>) already showed in 2011 that halting agricultural expansion and closing yield gaps instead should be a priority for future land-use planning.

We thank the reviewer for drawing out attention to these publications. The sentence on line 344 refers to biodiversity effects of land use change and therefore, it is not a suitable section to cite these studies that focused on closing yield gaps or sustainable intensification. Instead, we cite Folberth et al. (2020), Schneider et al. (2022) and Beyer et al. (2022) on lines 324-325 in the new manuscript.

Other comments:

Line 42: The following paper calculates where land-use expansion pressure could increase for a future scenario: Schneider et al. (2024) <https://doi.org/10.1038/s41893-024-01410-x>. This could be cited here and is maybe also interesting for a follow-up study.

We thank the reviewer for the suggested paper. While the study provides a scenario of where land-use change pressure could increase, this is based on only one model and one approach. The statement on line 42 (now line 40) refers to the uncertainty of where land-use pressure will act and to support that, we cite an analysis using 43 simulations of 11 land-use change models. Therefore, the suggested paper does not directly support the indicated statement.

Line 350-357: Primary vegetation in modified landscapes may hold threatened species. This should be discussed, since none of your metrics is considering this explicitly.

We thank the reviewer for raising this great point. We have now addressed this issue on line 245.

Reviewer #3:

Remarks to the Author:

I appreciate the hard work of the authors to revise the paper. I have read the rebuttal letter and the revised version of the paper. My major concern was about the novelty of the results, which is now much clearer from the replies. I still feel it should be more clearly mentioned in the paper, but that may be too pushy, at least in the abstract.

We thank the reviewer for the positive remarks. We have reviewed the paper and made some further adjustments. Moreover, we have edited the abstract with this comment in mind. We hope that the novelty of the results is now clearer.

Regarding the application of yield as indicator and not agrochemical use, my concern are still valid, but I accept that the authors found yield more fitting to their study.

We thank the reviewer for emphasising this aspect and we acknowledge the concerns as valid. As this is already mentioned in the limitations paragraph, we did not add anything else in the text at this time.

I still think homogenisation is not the best measure of community composition, as it measures only one character, the range size of species. I acknowledge, however, that this is an attempt to involve composition, which is usually neglected as indicator, although usually more sensitive than richness or abundance.

We thank the reviewer for emphasising this aspect. We have now modified the paragraph in the introduction dealing with homogenisation to make it clearer that range size is just one of the potential metrics that can be used to measure biotic homogenisation and community composition. We have also justified why range size is a useful trait from the point of view of conservation through its association with extinction risk. (lines 69-80)

On the 30% - 70% share Andrén's seminal paper is cited. That paper has limited relevance as it go for birds and mammals, and on a limited range of habitats and regions. I may add the concept of Tsharntke on simple and complex landscapes. There are more or less similar percentages, but linked to agricultural landscapes.

We thank the reviewer for pointing out this limitation in our justification. We have now further expanded the justification for this choice by citing Tsharntke et al. (2005) and Shennan-Farpón et al (2021) on the relevance of these thresholds in agricultural landscapes and tropical forests, respectively (line 100-102).